# Both Semantics and Reconstruction Matter: Making Representation Encoders Ready for Text-to-Image Generation and Editing

Shilong Zhang [1]   He Zhang [2]   Zhifei Zhang [2]   Chongjian GE [2]   Shuchen Xue [3]   Shaoteng Liu [2]   Mengwei Ren [2]
Soo Ye Kim [2]   Yuqian Zhou [2]   Qing Liu [2]   Daniil Pakhomov [2]   Kai Zhang [2]   Zhe Lin [2]   Ping Luo [1]

## Abstract

Modern Latent Diffusion Models (LDMs) typically operate in low-level Variational Autoencoder (VAE) latent spaces that are primarily optimized for pixel-level reconstruction. To unify vision generation and understanding, a burgeoning trend is to adopt high-dimensional features from representation encoders as generative latents. However, we empirically identify two fundamental obstacles in this paradigm: (1) the discriminative feature space lacks compact regularization, making diffusion models prone to off-manifold latents that lead to inaccurate object structures; and (2) the encoder's inherently weak pixel-level reconstruction hinders the generator from learning accurate fine-grained geometry and texture. In this paper, we propose a systematic framework to adapt understanding-oriented encoder features for generative tasks. We introduce a semantic–pixel reconstruction objective to regularize the latent space, enabling the compression of both semantic information and fine-grained details into a highly compact representation (96 channels with $16\times$ spatial downsampling). This design allows the latent space to remain semantically rich while achieving state-of-the-art image reconstruction, and keeps it compact enough for accurate generation. Leveraging this representation, we design a unified text-to-image (T2I) and image editing model. Across diverse generation spaces, our approach achieves state-of-the-art reconstruction, faster convergence, and substantial gains in both T2I and editing tasks, demonstrating that representation encoders can be effectively adapted into robust generative components.

[1]The University of Hong Kong [2]Adobe Research [3]University of Chinese Academy of Sciences. Correspondence to: Ping Luo <pluo@cs.hku.hk>.

*Proceedings of the $43^{rd}$ International Conference on Machine Learning*, Seoul, South Korea. PMLR 306, 2026. Copyright 2026 by the author(s).

## 1. Introduction

Representation encoders trained via self-supervision (Caron et al., 2021; Oquab et al., 2023; Siméoni et al., 2025; He et al., 2020; 2022) or contrastive learning (Radford et al., 2021; Tschannen et al., 2025; Bolya et al., 2025) have established themselves as the cornerstone of visual understanding. They produce highly discriminative, semantic-rich features that generalize exceptionally well, enabling efficient adaptation to downstream tasks with limited data (Liu et al., 2023; 2024a). From dense prediction tasks to complex reasoning in Large Vision Language Models, these encoders have become the universal bedrock of visual analysis. Yet, despite this pervasive dominance, these powerful representations have yet to conquer the generative domain.

Instead, state-of-the-art generative systems predominantly rely on Variational Autoencoder (VAE) (Kingma & Welling, 2013), which operate on low-level, compact latents trained with a pixel reconstruction objective. These VAE latents lack the high-level semantic structure of representation encoders, forcing diffusion models to learn visual concepts from scratch and necessitating massive computational resources (Esser et al., 2024). To bridge this divide and achieve the long-sought goal of unifying perception and generation, a natural question arises: *can we migrate the generative modeling space from VAE latents to the representation-encoder space, enabling diffusion models to directly benefit from the representation encoders' discriminative and semantically structured features?*

Recent work, Representation Autoencoder (RAE) (Zheng et al., 2025), offers a pioneering answer to this question. By redesigning the DiT architecture to handle high-dimensional features, it successfully enables generation within the representation space, achieving impressive results on the class-conditional ImageNet benchmark (Russakovsky et al., 2015). However, the efficacy of this paradigm does not easily translate to open-world applications. When extended to practical text-to-image synthesis and complex instruction-based editing tasks, RAE exhibits significant performance limitations compared to mature VAE-based baselines, as highlighted in Figure 1.

To uncover the root causes of this degradation, we analyze

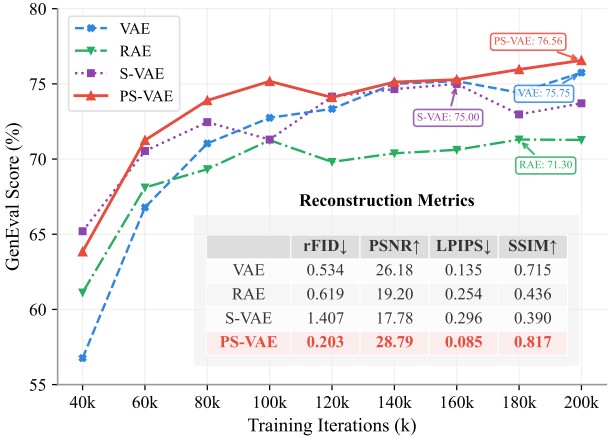

**Reconstruction Metrics**

| | rFID↓ | PSNR↑ | LPIPS↓ | SSIM↑ |
|---|---|---|---|---|
| VAE | 0.534 | 26.18 | 0.135 | 0.715 |
| RAE | 0.619 | 19.20 | 0.254 | 0.436 |
| S-VAE | 1.407 | 17.78 | 0.296 | 0.390 |
| **PS-VAE** | **0.203** | **28.79** | **0.085** | **0.817** |

*Figure 1.* **Reconstruction and generation performance across different generation spaces.** Compared to vanilla `VAE`, `RAE` improves generation coverage speed but quickly saturates due to its unconstrained semantic space and weak reconstruction. To address this, we project RAE features into a compact 96-channel latent space with a semantic reconstruction objective, forming `S-VAE`, which mitigates off-manifold issues and improves generation performance. Finally, `PS-VAE` further augments the semantic latent space with pixel-level reconstruction, enriching structural and texture details and achieving superior performance in both reconstruction and generation.

the behavior of the representation space both experimentally and theoretically in Section 3.2, identifying two key issues: insufficient compact regularization of representation features, leading to off-manifold latent generation, together with weak pixel-level reconstruction, which prevents the generator from learning accurate geometry and texture.

The first issue is that generation over representation features is performed in an unconstrained space without compact regularization, leading to a mismatch between the high dimensionality of representation features and their much lower intrinsic information content. Training on such a redundant high-dimensional space makes the diffusion model prone to producing off-manifold latents[1], ultimately leading to inaccurate and structurally distorted objects. We verify this phenomenon by visualizing off-manifold outliers through a toy fitting experiment and a theoretical analysis (Section 3.2 and Figure 3). This observation motivates us to regularize the generative space: we propose S-VAE, which maps the frozen representation features into a compact, KL-regularized latent space (Rombach et al., 2022) via a semantic autoencoder. This constraint effectively eliminates off-manifold outliers, ensuring that generated latents remain within the valid decoding manifold and thereby improving generation performance (as shown by the purple line in Figure 1).

The second issue arises from the training objective of representation encoders, which focuses on producing sufficiently discriminative features for understanding rather than pre-

---

[1]We define "off-manifold" latents as features falling into undefined/OOD regions where image decoding becomes unreliable.

serving detailed structure and fine-grained visual information required for generation. Consequently, even within the regularized `S-VAE` space, the model struggles to synthesize realistic fine-grained textures and precise small-scale structures. To address this, we unfreeze the encoder and jointly optimize it with a pixel-level reconstruction loss on the input image and a semantic reconstruction loss defined on the outputs of the original frozen pretrained encoder. This encourages the encoder to maintain fine-grained details during the computation of strong semantic representations, yielding our final Pixel–Semantic VAE (`PS-VAE`, Figure 1).

Specifically, we instantiate `PS-VAE` with a 96-channel latent design based on DINOv2 (Oquab et al., 2023). Compared to vanilla VAEs such as MAR-VAE (Li et al., 2024), this architecture achieves state-of-the-art reconstruction quality, significantly improving rFID (0.534 → 0.203), PSNR (26.18 → 28.79), and SSIM (0.715 → 0.817). It also outperforms MAR-VAE in text-to-image generation, exhibiting faster convergence and superior final performance (GenEval (Ghosh et al., 2023): 75.8 → 76.6; DPG-Bench (Hu et al., 2024): 83.2 → 83.6). Most notably, on the challenging instruction-based image editing task—requiring both accurate image understanding and faithful instruction execution—`PS-VAE` delivers a substantial improvement, boosting the editing reward from 0.06 to 0.22. We also validate our method on SigLIP2 (Tschannen et al., 2025), which is used in Bagel (Deng et al., 2025), observing consistent generation behavior. Importantly, the fine-tuned encoder retains strong understanding ability without any LLM fine-tuning, highlighting the potential of our approach for unifying the encoder for both understanding and generation.

## 2. Related Work

**Representation Encoders for Visual Understanding** Pretrained representation encoders (Caron et al., 2021; Oquab et al., 2023; Siméoni et al., 2025; Zhai et al., 2023; Tschannen et al., 2025; Bolya et al., 2025) are central to modern visual understanding, supporting tasks including classification, detection, segmentation, and vision–language modeling (Liu et al., 2023; Tong et al., 2024). These encoders are typically trained via self-supervised learning (Chen et al., 2020; He et al., 2020; Grill et al., 2020; Caron et al., 2021; Oquab et al., 2023; Siméoni et al., 2025) or image–text contrastive learning (Radford et al., 2021; Zhai et al., 2023; Bolya et al., 2025). Despite their success in visual understanding, their application to image generation remains relatively underexplored and immature.

**VAEs for Visual Generation** VAEs (Kingma & Welling, 2013) are core components of latent diffusion models (Rombach et al., 2022), enabling efficient high-resolution generation. However, VAEs trained primarily with pixel-level reconstruction objectives tend to emphasize low-level struc-

ture while lacking high-level semantics, forcing diffusion models to relearn visual concepts from scratch. Recent studies show that latent-space topology and regularization are crucial for generation quality (Kouzelis et al., 2025a; Skorokhodov et al., 2025; Yao et al., 2025; Leng et al., 2025; Lu et al., 2025; Lee et al., 2025; Kouzelis et al., 2025b); these improvements are often associated with stronger semantic structure in the latent space. Building on this insight, we start directly from a representation encoder and make it compact while improving pixel-level reconstruction, thereby making it suitable for both image generation and editing.

**Unifying Feature Spaces for Generation and Understanding** Prior efforts to unify discriminative and generative representations follow two main directions. One aligns VAE latents with representation encoders via semantic regularization (Yao et al., 2025; Xu et al., 2025). The other constructs generative spaces directly from representation encoders (Chen et al., 2025a; Lu et al., 2025; Yue et al., 2025; Zheng et al., 2025; Shi et al., 2025), often without enforcing compactness or full semantic preservation (Chen et al., 2025a; Lu et al., 2025; Yue et al., 2025). Closely related works such as SVG (Shi et al., 2025) and RAE (Zheng et al., 2025) diffuse over raw, high-dimensional representation features, but lack compact regularization, leading to off-manifold generation and weak pixel-level reconstruction that limits geometric and textural fidelity. The Concurrent work FAE (Gao et al., 2025) is more aligned with our semantic autoencoding stage; its weak reconstruction can limit fine details and texture quality in text-to-image generation, as well as its applicability to image editing. Approaches (Ma et al., 2025; Song et al., 2025; Lin et al., 2025; Han et al., 2025; Chen et al., 2025b; Qu et al., 2025) aim to achieve unified modeling in an autoregressive manner; however, their performance in text-to-image generation still lags significantly behind diffusion-based methods.

# 3. Method

In this section, we first introduce a Deep-Fusion architecture for text-to-image generation and image editing, which serves as a fair benchmarking framework for comparing different generative latent spaces. We then analyze the reconstruction and generation behavior of the Representation Autoencoder (RAE) (Zheng et al., 2025). Through both theoretical and empirical analysis, we show that its unconstrained feature space induces off-manifold latent generation during diffusion, producing features outside the training support of the pixel decoder. Moreover, RAE's poor reconstruction fidelity prevents the generative model from learning accurate object structures and fine-grained textures, limiting its effectiveness for high-fidelity tasks such as instruction-based editing. To address these issues, we introduce a step-wise strategy that prepares representation encoders for generation by mapping both pixels and representations into a unified,

compact latent space. Unless otherwise specified, we follow the settings of RAE (Zheng et al., 2025) and use a DINOv2-B (Oquab et al., 2023) encoder for feature extraction.

## 3.1. Generation Architecture

Unified models for generation and understanding have recently attracted increasing attention. With strong semantic representations and high-fidelity reconstruction, `PS-VAE` is well-suited to serve as a potential unified encoder in such frameworks, motivating our adoption of a deep-fusion generation architecture. We compare several deep-fusion architectures (LlamaFusion (Shi et al., 2024), Bagel (Deng et al., 2025), and Transfusion (Zhou et al., 2025)) for text-to-image generation and find that Transfusion offers the best parameter–performance trade-off. We further adopt a wide DDT head (Zheng et al., 2025; Wang et al., 2025), which consistently improves generation across multiple VAE latent spaces and reduces sensitivity to channel dimensionality (we explain this in Section 3.2). All architectural ablations and discussion on how this framework supports both generation and editing are deferred to Section A.1.

## 3.2. Analysis of RAE

**Comparison: RAE *vs.* VAE** We analyze the reconstruction and generation behavior of a representation autoencoder (RAE (Zheng et al., 2025), using DINOv2 features with a pixel decoder) by comparing it with a standard VAE baseline (Li et al., 2024) under the same $16 \times 16$ spatial compression. As shown in Figure 2(a) and Table 3, RAE exhibits substantially worse reconstruction quality than VAE, with noticeably lower SSIM and PSNR and visible artifacts in regions such as faces and text. This is unsurprising, as the DINOv2 encoder is trained with a purely discriminative objective and does not explicitly optimize for reconstruction. We further evaluate their impact on generation using both text-to-image and image editing tasks (all equipped with a wide DDT head (Wang et al., 2025; Zheng et al., 2025)). Despite faster coverage in text-to-image generation enabled by its strong semantic feature space (see Figure 1), RAE suffers from severe structural and texture artifacts (see Figure 2(c)) and significantly underperforms VAE on benchmarks such as GenEval (Ghosh et al., 2023). For image editing, RAE shows stronger prompt-following in semantically driven edits (see Figure 2(b)), but its poor reconstruction quality limits fine-grained detail preservation.

While most RAE behaviors are expected (e.g., inferior reconstruction, faster coverage, and better prompt-following in editing), *RAE exhibits disproportionately severe artifacts during generation, far beyond what its reconstruction quality would suggest.* Notably, this issue is largely overlooked in the original RAE class-to-image evaluations, where FID (Heusel et al., 2017) is the primary metric (see Section A.3).

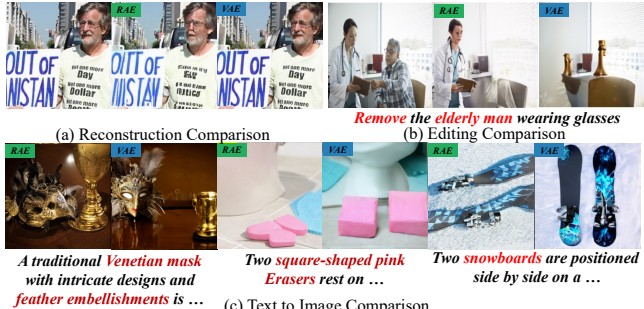

(a) Reconstruction Comparison

(b) Editing Comparison

*Remove the elderly man wearing glasses*

*A traditional Venetian mask with intricate designs and feather embellishments is …*

*Two square-shaped pink Erasers rest on …*

*Two snowboards are positioned side by side on a …*

(c) Text to Image Comparison

*Figure 2.* **Visualization comparison between RAE and VAE.** (a) RAE shows a noticeable gap in reconstruction performance compared to VAE. Benefiting from its rich semantic representation, RAE demonstrates stronger prompt-following ability in image editing tasks that require understanding the input image (b). However, its poor reconstruction quality limits practical usability, as it fails to preserve fine-grained and consistent details from the input image. Counterintuitively, in text-to-image generation, RAE exhibits severe structural and texture artifacts and substantially lags behind VAE (c), with a performance gap far larger than that observed in reconstruction.

**Analysis of Off-Manifold Behavior in RAE** The structural and texture artifacts in RAE far exceed its reconstruction errors, indicating a fundamental failure of the generator caused by adopting representation encoder features as the generative space. This failure stems from the fact that such representation features do not form a compact and well-behaved generative manifold. Consequently, diffusion models trained in the high-dimensional RAE feature space are difficult to optimize and struggle to stay close to the data manifold during generation, frequently producing off-manifold samples that fall outside the training distribution of the pixel decoder and lead to degraded decoding.

We provide a detailed comparison between diffusion learning in unconstrained high-dimensional ambient spaces and learning on low-dimensional intrinsic manifolds, supporting our diagnosis. We model the generative dynamics of an $h$-dimensional feature space containing an $l$-dimensional manifold ($h > l$). Let $z \in \mathbb{R}^l$ denote the latent data and $x = Qz \in \mathbb{R}^h$ denote the observed data, where $Q \in \mathbb{R}^{h \times l}$ is a column-orthonormal mapping ($Q^\top Q = I_l$). The forward diffusion processes for $x$ and $z$ are coupled: $x_t = (1-t)x_0 + t\epsilon_h$ implies that the projected variable $z_t = Q^\top x_t$ follows $z_t = (1-t)z_0 + t\epsilon_l$, where $\epsilon_l = Q^\top \epsilon_h$. Beyond the coupling of the forward processes, the optimal denoising objectives are strictly related. We denote the optimal velocity estimators for the intrinsic and embedded processes as $v_{z,\theta}$ and $v_{x,\theta}$, respectively. These are defined as the expected velocity targets given the noisy states:

$$v_{z,\theta}(z_t) = \mathbb{E}[\epsilon_l - z_0 | z_t], \quad v_{x,\theta}(x_t) = \mathbb{E}[\epsilon_h - x_0 | x_t]$$

By projecting the signal in the embedded high dimensions onto the data manifold and its orthogonal complement, we can express the high-dimensional estimator $v_{x,\theta}$ purely in terms of the low-dimensional estimator $v_{z,\theta}$ plus a residual

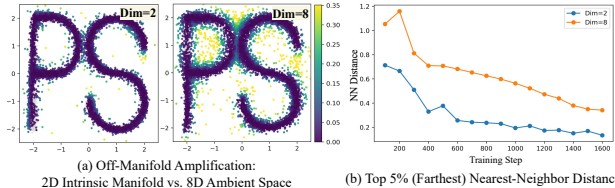

(a) Off-Manifold Amplification: 2D Intrinsic Manifold vs. 8D Ambient Space

(b) Top 5% (Farthest) Nearest-Neighbor Distance

*Figure 3.* **Off-manifold behavior varies significantly with feature dimensionality.** We construct a 2D 'PS'-shaped distribution and embed it into an 8D ambient space, yielding two learning settings with intrinsic dimension 2 and ambient dimension 8. **(a)** The 8D setting produces substantially more off-manifold samples than the intrinsic 2D space. **(b)** We measure the mean nearest-neighbor distance of the top 5% tail samples and observe that samples generated in 8D deviate much farther from the data manifold, indicating stronger off-manifold drift.

term:

$$v_{x,\theta}(x_t) = Q v_{z,\theta}(Q^\top x_t) + \frac{1}{t}(I - QQ^\top)x_t \qquad (1)$$

Equation 1 reveals a fundamental disparity in learning difficulty. The first term represents the generative flow along the intrinsic manifold. While this component is intrinsically low-dimensional, the model operating in the ambient space must implicitly learn the projection ($Q^\top$) and embedding ($Q$) operations from noisy inputs to recover it. This introduces a substantial manifold discovery burden that is avoided by diffusion in the latent space.

The second term contains pure Gaussian noise in the orthogonal subspace. *Since Gaussian noise is full-rank and incompressible, the network is forced to learn an identity-like mapping to transmit noise, leading to substantial waste of model capacity when learning in a high-dimensional ambient space.* [2]

We validate our analysis with a toy experiment that embeds a 2D "PS"-shaped distribution into an 8D ambient space via a linear isometry. Identical diffusion models are trained in either the intrinsic 2D space or the 8D ambient space, with 8D samples projected back for evaluation. As shown in Figure 3, training in the ambient space converges more slowly and produces pronounced off-manifold samples, demonstrating that unconstrained high-dimensional representations destabilize diffusion training despite identical intrinsic geometry. Further details are provided in Section A.4.

### 3.3. Make Representation Encoders Ready

Building on our analysis of RAE (Zheng et al., 2025), we first address the off-manifold problem, identified as the key limitation. Subsequently, we enhance reconstruction fidelity to improve the text-to-image performance and enable detail-sensitive applications such as image editing.

---

[2]This explains why RAE (Zheng et al., 2025) struggles to fit even a single sample when the model dimensionality is smaller than the input dimensionality, and clarifies the effectiveness of the wide DDT-Head (Wang et al., 2025; Zheng et al., 2025), which bypasses this burden via a long skip connection.

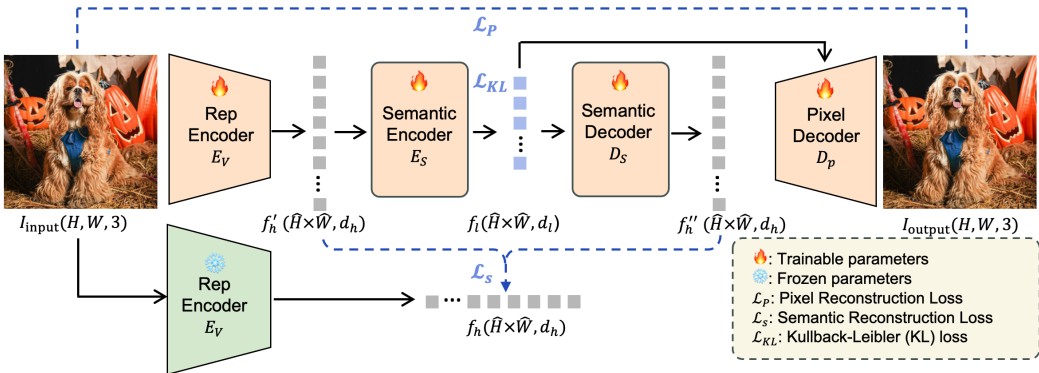

*Figure 4.* **Compact latent space construction for preserving semantic structure and fine-grained details** We first regularize the unconstrained representation-encoder feature space by freezing the encoder and training a semantic VAE using only the $\mathcal{L}_s$ and $\mathcal{L}_{kl}$; during this stage, the pixel decoder is trained on the detached semantic latent with pixel reconstruction loss $\mathcal{L}_P$. After semantic reconstruction converges, we unfreeze all components and allow the pixel decoder to backpropagate the gradient into the encoder, ensuring that the representation encoder captures fine-grained details of the input image.

The framework of `PS-VAE` is illustrated in Figure 4. Given an input image $I_{\text{input}} \in \mathbb{R}^{H \times W \times 3}$, we first extract a semantic feature map $f'_h \in \mathbb{R}^{\hat{H} \times \hat{W} \times d_h}$ using a pretrained representation encoder, *e.g.* DINOv2-B (Oquab et al., 2023). As discussed in Section 3.2, $f'_h$ is a high-dimensional, unconstrained representation, where its dimension $d_h = 768$.

To address the off-manifold problem, we introduce a semantic VAE (S-VAE) that maps the high-dimensional unconstrained feature space $f'_h$ to a compact latent space $f_l \in \mathbb{R}^{\hat{H} \times \hat{W} \times d_l}$ via an encoder $E_S(d_l = 96)$, yielding a much more compact representation than the original 768-dimensional features. The semantic decoder $D_S$ is adopted to reconstruct the latent back to the original feature $f''_h$. The semantic encoder and decoder share a symmetric design with only three Transformer blocks inherited from the representation encoder and an MLP projection layer for dimensionality adjustment, introducing limited overhead compared with downstream diffusion training. At $256 \times 256$ resolution, `PS-VAE` requires roughly 27.7G FLOPs, lower than VAVAE (Yao et al., 2025) ($\sim$69.2G) and about one-fifth of Flux-VAE (Labs, 2024) ($\sim$138G).

The semantic encoder $E_S$ and decoder $D_S$ are optimized with a semantic reconstruction loss $\mathcal{L}_S$, which combines an $\ell_2$ loss, a cosine similarity loss, and a matrix similarity loss following (Yao et al., 2025) to ensure that the reconstructed feature $f''_h$ closely matches the original representation $f'_h$. The latent is further regularized by a Kullback–Leibler divergence loss $\mathcal{L}_{KL}$ following (Rombach et al., 2022) to further enforce a compact and well-behaved latent distribution. The representation encoder is frozen during this stage. *More details of the loss components can be found in Section A.5.*

For the reconstruction evaluation of S-VAE, we additionally train a pixel decoder that reconstructs the output image $I_{\text{output}}$ from the detached semantic latent $f_l.\text{detach}()$ via the pixel reconstruction loss $\mathcal{L}_P$ (following (Rombach et al.,

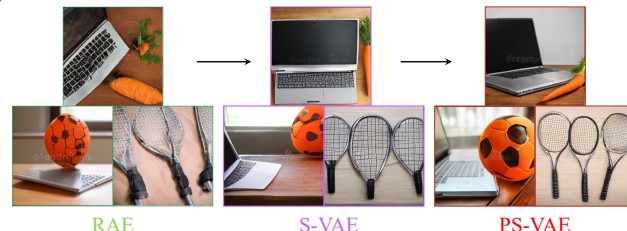

*Figure 5.* **Visual comparison of generated examples across progressively improved latent spaces** (RAE → S-VAE → PS-VAE). Artifacts are gradually reduced, with step-by-step improvements in texture and structure.

2022)). A diffusion model is further trained on (S-VAE) latent space. As shown in Figure 5 and Table 3, both visual quality and quantitative results are substantially improved, despite a slight performance drop in reconstruction fidelity. This result confirms that the primary limitation lies in the off-manifold issue rather than reconstruction quality.

To enhance image reconstruction without compromising the semantic structure of the latent space, we unfreeze the representation encoder during pixel decoder training. By removing the *detach* operation in $f_l$, we enable $\mathcal{L}_P$ gradients to propagate from the pixel decoder back to the representation encoder. To preserve the pretrained semantic representations during this optimization, we enforce a semantic reconstruction loss on $f'_h$ and $f''_h$ relative to the original encoder, while retaining the $\mathcal{L}_{KL}$ to ensure that the latent space remains compact. After this training stage, we obtain our Pixel–Semantic VAE (PS-VAE). The overall loss components and the inputs to each loss term are shown in Equation (2). Unless otherwise specified, we set $\lambda_S = 0.1$, $\lambda_P = 1.0$, and $\beta = 1 \times 10^{-6}$. *More details of the loss components can be found in Section A.5.*

$$\mathcal{L} = \lambda_S \mathcal{L}_S(f_h, f'_h, f''_h) + \lambda_P \mathcal{L}_P(I_i, I_o) + \beta \mathcal{L}_{KL}(f_l). \quad (2)$$

As demonstrated in Table 3, this strategy significantly improves the reconstruction quality of the representation encoder while preserving its semantic structure. This enables the generation model to learn fine-grained geometry and

texture (as shown in Figure 5), while the well-preserved semantics ensure fast coverage of text-to-image pretraining and strong instruction-following ability for image editing (as shown in Figure 6). Consequently, `PS-VAE` enables representation encoders to surpass VAE-based models across reconstruction, text-to-image generation, and image editing.

# 4. Experiments

Due to space limitations, we briefly summarize the training and inference procedures and the evaluation protocols for reconstruction, text-to-image generation, and image editing; *full details are provided in Section A.2*. We then present performance results across varying feature spaces to demonstrate the effectiveness of `PS-VAE`. Subsequently, we analyze the scaling behavior of our 96- and 32-channel variants, showing that larger generation models effectively leverage the rich semantic and pixel-level details preserved in higher-channel latent spaces. Finally, we extend our framework by replacing the DINOv2 (Oquab et al., 2023) encoder with SigLIP2 (Tschannen et al., 2025), which not only demonstrates the generality of our pipeline but also highlights its potential as a unified encoder for both visual understanding and generative modeling.

## 4.1. Training and Evaluation Details

We briefly summarize task-specific training and evaluation details below. *Full details are provided in Section A.2.*

*Reconstruction* `PS-VAE` is trained on ImageNet-1K (Russakovsky et al., 2015) using a two-stage strategy (`S-VAE` followed by `PS-VAE`). We evaluate the reconstruction performance on the ImageNet-1K validation set.

*Text-to-Image Generation* Text-to-image models are trained on CC12M-LLaVA-NeXT (Changpinyo et al., 2021; Emporium, 2024) and evaluated with GenEval (Ghosh et al., 2023) and DPG-Bench (Hu et al., 2024). We adopt a channel- and patch-aware timestep shift to ensure a consistent SNR across latent spaces for fair comparison.

*Instruction Editing* Editing models are initialized from text-to-image checkpoints, trained on OmniEdit (Wei et al., 2024), and evaluated with EditingReward (Wu et al., 2025).

## 4.2. Rec. & Gen. across Feature Spaces

As shown in Table 1, $PS-VAE_{96c}$ achieves the highest reconstruction quality among stride-16 VAEs, with Flux-VAE being the only method achieving higher reconstruction quality. Nevertheless, Flux-VAE performs substantially worse in text-to-image generation and image editing. In generation and editing tasks, both $PS-VAE_{32c}$ and $PS-VAE_{96c}$ significantly outperform RAE. Specifically, $PS-VAE_{32c}$ achieves top performance on DPG-Bench (Hu et al., 2024) and Editing Reward (Wu et al., 2025), ranking second on GenEval (Ghosh et al., 2023). Meanwhile, $PS-VAE_{96c}$ leads on GenEval and ranks second and third on DPG-

Bench and Editing Reward, respectively, maintaining a clear advantage over the RAE baseline. Moreover, benefiting from a well-constrained and semantically rich latent space, `PS-VAE` converges faster than RAE and other VAEs (see Figure 6). Furthermore, enhanced detail fidelity enables `PS-VAE` to surpass standard VAE at final performance, highlighting the distinct advantage of the pixel–semantic constrained latent space.

Meanwhile, we observe in Table 1 and Figure 12 that VAEs trained solely on pixel reconstruction objectives (e.g., MAR-VAE and Flux-VAE) exhibit significantly lower prompt-following capabilities than models with semantically structured latent spaces, *e.g.* $PS-VAE_{32c}$, $PS-VAE_{96c}$, and VAVAE. We hypothesize that instruction-based editing involves two coupled components: interpreting the source image latent and generating the edited output following the prompt. Consequently, a semantically organized latent space facilitates source image interpretation and improves instruction adherence. We also observe from the visual comparisons that RAE's editing performance is constrained by weak reconstruction capabilities: although it can localize the target regions and follow high-level instructions, the edited images often drift from the input appearance and lose identity-level details. By contrast, `PS-VAE` effectively combines a semantically organized representation with fine-grained detail preservation, retaining the strong instruction-following ability of RAE (see Figure 12.a) while achieving superior visual consistency in local regions such as facial features, clothing textures, and background details (see Figure 12.a,b,c).

## 4.3. Diffusion Scaling across PS-VAE Channels

As shown in Figure 6 and Table 1, both $PS-VAE_{32c}$ and $PS-VAE_{96c}$ achieve state-of-the-art generation performance. While $PS-VAE_{96c}$ provides higher reconstruction quality, its generation performance slightly underperforms that of $PS-VAE_{32c}$ under limited model capacity, indicating that modeling richer fine-grained details benefits from a larger generative backbone, in line with prior observations (Esser et al., 2024; Yao et al., 2025).

To verify this, we scale the diffusion backbone from Qwen-0.5B to Qwen-1.5B (Bai et al., 2023). As shown in Figure 7, $PS-VAE_{96c}$ exhibits consistent improvements across all generation and editing benchmarks, whereas $PS-VAE_{32c}$ shows diminishing returns or even degradation. This indicates that higher-channel `PS-VAE` scale better and achieve higher performance ceilings with larger models.

Finally, we fine-tune Qwen-3B (Bai et al., 2023) with $PS-VAE_{96c}$ for 600k iterations, including 200k iterations under the same training setting and an additional 400k iterations on a high-quality internal dataset. Leveraging the strong semantic representation and high-fidelity reconstruction of our latent space, the model produces images with

*Table 1.* **Comparison of reconstruction and generation performance.** The best results are shown in **bold** and the second-best are underlined. Flux-VAE (stride 8) is listed for reference, and all other results correspond to the feature space with a stride 16.

| Method | rFID ($\downarrow$) | PSNR ($\uparrow$) | LPIPS ($\downarrow$) | SSIM ($\uparrow$) | GenEval ($\uparrow$) | DPG-Bench ($\uparrow$) | Editing Reward ($\uparrow$) |
|---|---|---|---|---|---|---|---|
| Flux-VAE (Labs, 2024) | **0.175** | **32.86** | **0.044** | **0.912** | 68.04 | 78.98 | -0.271 |
| MAR-VAE (Li et al., 2024) | 0.534 | 26.18 | 0.135 | 0.715 | 75.75 | 83.19 | 0.056 |
| VAVAE (Yao et al., 2025) | 0.279 | 27.71 | 0.097 | 0.779 | 76.16 | 82.45 | 0.227 |
| RAE (Zheng et al., 2025) | 0.619 | 19.20 | 0.254 | 0.436 | 71.27 | 81.72 | 0.059 |
| **PS-VAE**$_{32c}$ | 0.584 | 24.53 | 0.168 | 0.662 | 76.22 | **84.25** | **0.274** |
| **PS-VAE**$_{96c}$ | 0.203 | 28.79 | 0.085 | 0.817 | **76.56** | 83.62 | 0.222 |

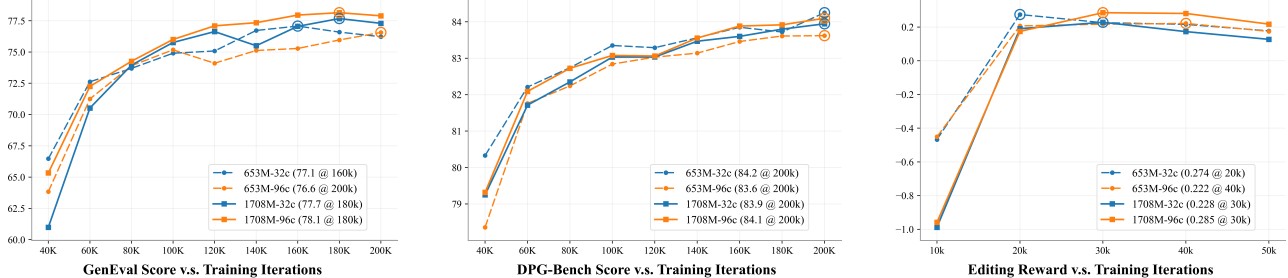

*Figure 6.* **Coverage curves for generation and editing across feature spaces.** Strong semantics of PS-VAE enable faster convergence in text-to-image and better instruction following in image editing, while high-fidelity reconstruction preserves the upper bounds of both generation and editing by yielding more accurate textures and finer-grained structures in generation, and by further improving detail consistency in image editing.

*Figure 7.* Scaling behavior of 653M (dashed) and 1708M (solid) models under different PS-VAE channel sizes (32c/96c) on (a) GenEval, (b) DPG-Bench, and (c) Editing Reward. PS-VAE$_{96c}$ exhibits consistent improvements across all tasks, with GenEval rising from 76.56 to 78.14, DPG-Bench from 83.62 to 84.09, and Editing Reward increasing significantly from 0.222 to 0.285. In contrast, PS-VAE$_{32c}$ demonstrates diminishing performance, showing only marginal gains on GenEval (77.07 to 77.67) and slight degradation on both DPG-Bench (84.25 to 84.10) and Editing Reward (0.274 to 0.228). These results indicate that higher-channel latent spaces possess superior scaling properties and higher upper bounds when paired with a larger generative model. Investigating the correspondence between high-channel latent spaces and large-scale generation backbones represents a promising direction for future research.

accurate text rendering, high-quality portraits, and flexible compositions of complex concepts (see Figure 8). Notably, these results are achieved while training solely at $256 \times 256$ resolution, and we expect further gains when extending to higher resolutions, which we leave for future exploration.

### 4.4. PS-VAE with SigLIP2: Toward Unified Understanding and Generation

While SigLIP2 (Tschannen et al., 2025) is widely used for multimodal understanding, we investigate whether PS-VAE enables SigLIP2 to serve as a unified encoder for both understanding and generation. We adopt the SigLIP2-so400m/14 encoder from Bagel (Deng et al., 2025) and adjust only the

*Table 2.* **Comparison between DINOv2 and SigLIP2 backbones.**

| Method | rFID$\downarrow$ | PSNR$\uparrow$ | LPIPS$\downarrow$ | SSIM$\uparrow$ | GenEval$\uparrow$ | DPG$\uparrow$ | ER$\uparrow$ |
|---|---|---|---|---|---|---|---|
| **PS-VAE**$_{96c}$ (DINOv2-B) | 0.203 | 28.79 | 0.085 | 0.817 | 76.56 | 83.62 | 0.222 |
| **PS-VAE**$_{96c}$ (SigLIP2-so400m/14) | 0.222 | 28.14 | 0.096 | 0.795 | 77.14 | 83.33 | 0.183 |

$\mathcal{L}_S : \mathcal{L}_P$ ratio from 0.1:1 to 0.05:1 in the PS-VAE stage to accommodate its more abstract representations.

**Generation Performance** As shown in Table 2, PS-VAE$_{96c}$(SigLIP2) achieves generation performance comparable to PS-VAE$_{96c}$(DINOv2), with similar reconstruction quality and competitive results across GenEval, DPG-Bench, and Editing Reward. These results demonstrate the robustness and transferability of PS-VAE.

**Understanding Performance** To assess whether pixel-level

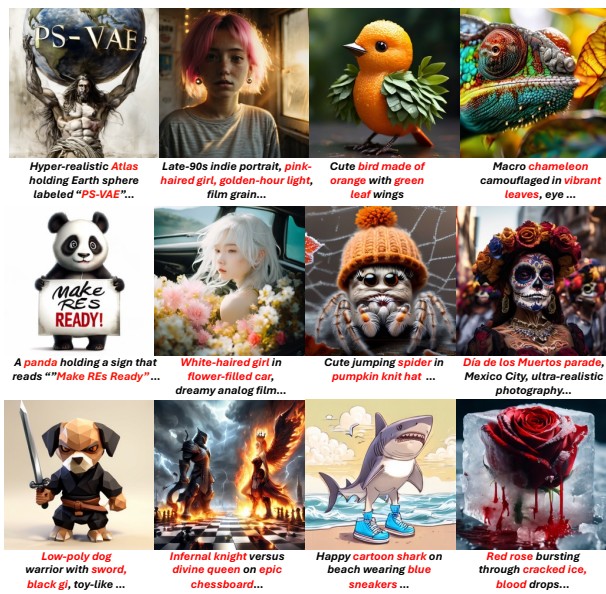

*Figure 8.* **Text-to-image generation examples using PS-VAE**$_{96c}$. Despite being trained only at $256\times256$ resolution, the semantically structured and detail-preserving latent space enables the generator to accurately follow complex text prompts, yielding images with correct structures, fine-grained textures, precise text rendering, realistic portraits, and flexible compositions of abstract concepts.

reconstruction degrades semantic representations, we replace the original encoder in Bagel with our fine-tuned SigLIP2 encoder while freezing the LLM. On standard benchmarks, we observe only marginal drops (e.g., MME-P (Fu et al., 2023) from 1685 to 1652 and MMBench (Liu et al., 2024b) from 85.0 to 84.7). Notably, these results are achieved without fine-tuning the LLM. Joint training with Bagel may further improve performance beyond the original baseline, which we leave for future work, as the fine-tuned encoder retains rich visual details while maintaining a well-structured semantic representation.

## 5. Ablation Study

### 5.1. Evolution from RAE to PS-VAE

**RAE → S-VAE** By introducing semantic reconstruction to map RAE features into a compact space, S-VAE substantially improves generation and editing performance over RAE (see Table 3), despite degraded reconstruction quality. This indicates that off-manifold latent samples are a key limitation of RAE during generation.

**S-VAE → PS-VAE** By additionally introducing pixel reconstruction to S-VAE, PS-VAE recovers high-frequency visual details without sacrificing semantic coherence. The 96-channel variant achieves strong reconstruction quality and surpasses MAR-VAE on GenEval, DPG-Bench, and Editing Reward, and nearly doubles the editing reward compared to S-VAE, demonstrating strong editing consistency.

**Semantic Structure Matters** Training a pixel-only recon-

*Table 3.* **From RAE to PS-VAE.** Making the representation compact is important for reducing off-manifold behavior and improving generation and editing (RAE → S-VAE). Adding pixel reconstruction while preserving semantic structure yields PS-VAE with the best overall performance. In contrast, a pixel-only reconstruction VAE (P-VAE) achieves stronger reconstruction but degrades semantic quality, reflected by lower linear probing (LP, top-1 / top-5), DPG-Bench, and editing reward (ER) scores.

| Method | rFID↓ | PSNR↑ | LPIPS↓ | SSIM↑ | LP↑ | GenEval↑ | DPG↑ | ER↑ |
|---|---|---|---|---|---|---|---|---|
| MAR-VAE | 0.534 | 26.18 | 0.135 | 0.715 | 5.4/15.1 | 75.7 | 83.2 | 0.06 |
| RAE | 0.619 | 19.20 | 0.254 | 0.436 | **83.0/96.6** | 71.3 | 81.7 | 0.06 |
| S-VAE | 1.407 | 17.78 | 0.296 | 0.390 | 81.1/95.7 | 73.7 | **83.6** | 0.12 |
| P-VAE | 0.398 | **29.81** | **0.073** | **0.850** | 11.8/26.7 | 75.2 | 82.1 | 0.04 |
| **PS-VAE** | **0.203** | 28.79 | 0.085 | 0.817 | 79.5/94.8 | **76.6** | 83.6 | **0.22** |

struction variant (P-VAE) causes semantic quality to regress to the level of MAR-VAE, leading to clear drops in DPG performance and editing reward. This confirms that the semantic representation of the latent space is essential for text alignment and instruction following.

### 5.2. Ablation on Latent Channel Dimensions

To identify the optimal channel dimension of PS-VAE, we search over channel sizes from 32 to 256. As shown in Figure 9, reconstruction performance saturates at 112 channels. As for generation performance, increasing latent dimensionality slows convergence (see Figure 9). However, final performance remains comparable between 32 and 96 channels on both GenEval and DPG-Bench. A turning point occurs at 112 channels, where the DPG-Bench score drops noticeably by approximately 0.6 points. This observation suggests that, under our training setup, the effective latent dimensionality required to jointly preserve semantic structure and pixel details is ∼96 channels. Increasing the channel number beyond this point degrades generation, likely due to overemphasizing high-frequency details.

### 5.3. Ablation on Encoder and Decoder Architectures

As shown in Table 4, projecting an unconstrained representation into a compact, KL-regularized latent space incurs non-trivial computational overhead. A shallow 2-layer MLP fails to preserve rich semantics in a 96-channel latent, leading to a clear drop in linear probing accuracy. Limited mapping capacity reduces the effective intrinsic dimensionality below 96, causing channel collapse under KL regularization and degrading generation performance. We also examine a symmetric architecture that feeds reconstructed semantic features directly into the pixel decoder. While this improves reconstruction quality, it degrades GenEval performance, likely due to gradient interference between semantic and pixel objectives along shared Transformer paths. A more careful re-balancing of the loss terms may alleviate this issue and is left for future work.

### 5.4. RAE-HD Induces Shortcuts

We attribute this failure to the inherent difficulty of constraining a high-dimensional latent space. Even with semantic-

| #C | rFID (↓) | PSNR (↑) | LPIPS (↓) | SSIM (↑) |
|---|---|---|---|---|
| 32 | 0.584 | 24.53 | 0.168 | 0.662 |
| 48 | 0.475 | 25.43 | 0.147 | 0.697 |
| 64 | 0.423 | 26.65 | 0.124 | 0.744 |
| 80 | 0.292 | 27.38 | 0.107 | 0.772 |
| 96 | 0.203 | 28.79 | 0.085 | 0.817 |
| 112 | 0.159 | **30.51** | **0.064** | **0.865** |
| 256 | **0.156** | 30.30 | 0.065 | 0.860 |

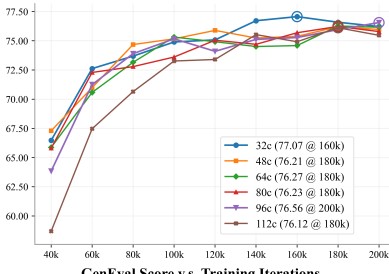
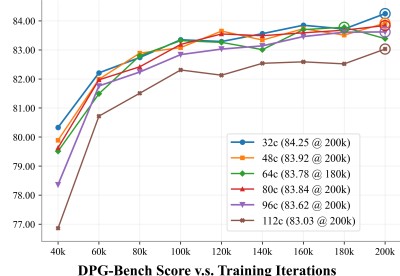

*Figure 9.* **Ablation on latent channel dimensions of `PS-VAE`.** *Left:* reconstruction metrics vs. channel dimensionality. *Middle/Right:* convergence on GenEval and DPG-Bench. Increasing channels slightly slows convergence; performance remains stable from 32c to 96c, but degrades beyond 96c, suggesting an intrinsic latent dimensionality of ∼96 channels under our training setup. Higher channels tend to emphasize high-frequency details, which may consume additional model capacity and interfere with semantic learning.

*Table 4.* **Comparison of `PS-VAE` design variants.** 2L-MLP S-VAE denotes an S-VAE in which both the semantic encoder and the semantic decoder are implemented as 2-layer MLPs. $D_{Pixel}$ on $D_{Semantic}$ indicates that the pixel decoder is attached to the semantic decoder features during the final detail-enrichment training stage.

*Table 5.* **High-dimensional enrichment causes shortcut reconstruction.** RAE-HD greatly improves reconstruction metrics but harms generation, indicating loss of semantic structure and the use of shortcut reconstruction.

| Method | rFID (↓) | PSNR (↑) | LPIPS (↓) | SSIM (↑) | Linear (↑) | GenEval (↑) | DPG (↑) |
|---|---|---|---|---|---|---|---|
| **PS-VAE** | 0.214 | 28.63 | 0.087 | 0.813 | 79.5 / 94.8 | 76.6 | 83.6 |
| 2L-MLP S-VAE | 0.205 | 28.94 | 0.082 | 0.812 | 44.5 / 64.5 | 70.3 | 83.1 |
| $D_{Pixel}$ on $D_{Semantic}$ | 0.193 | 29.64 | 0.077 | 0.840 | 80.4 / 95.4 | 74.4 | 83.6 |

| Method | rFID↓ | PSNR↑ | LPIPS↓ | SSIM↑ | GenEval↑ |
|---|---|---|---|---|---|
| RAE | 0.619 | 19.20 | 0.254 | 0.436 | 71.3 |
| RAE-HD | 0.193 | 33.10 | 0.048 | 0.916 | 60.2 |

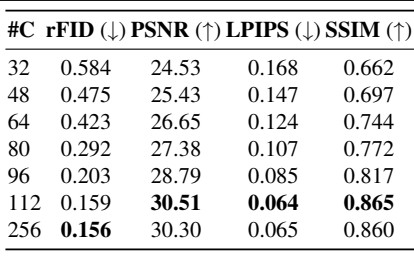
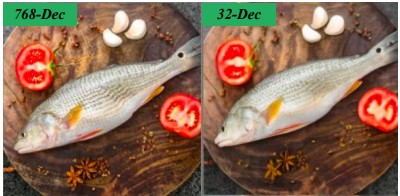

(a) Severe structural artifacts and incoherent textures

(b) Reconstruction shortcut behavior: comparable reconstruction with only 32 of 768 channels.

*Figure 10.* Directly enriching details in a high-dimensional space leads to severe generation artifacts (a). We further verify that this behavior arises from reconstruction shortcuts in high-dimensional feature spaces (b).

preserving losses, the model can exploit shortcut solutions by relying on a sparse subset of channels for reconstruction, without inducing meaningful changes in feature distances in high-dimensional spaces, where distance metrics tend to become less informative. We verify this shortcut behavior by showing that retraining a pixel decoder using only the 32 selected channels (out of 768) with the largest deviations between the fine-tuned encoder and the frozen DINOv2 features is sufficient to achieve strong reconstruction performance (as shown in Figure 10.b). This indicates that constraining detail enrichment to a compact, semantically regularized latent space is essential, thereby validating the core design of our `PS-VAE`.

An alternative strategy to improve pixel reconstruction involves training the pixel decoder directly on the original high-dimensional feature space, while maintaining the semantic reconstruction loss with a frozen DINOv2 encoder. As shown in Table 5, while this approach yields rapid improvements in reconstruction quality, it leads to a significant degradation in generation performance. The generated images exhibit severe structural artifacts and incoherent textures (as shown in Figure 10.a).

## 6. Conclusion

In this work, we show that powerful representation encoders, despite their strong discriminative capabilities, are not directly suitable as generative spaces due to unconstrained feature distributions and limited reconstruction fidelity. Through systematic analysis, we identify off-manifold generation and poor reconstruction as the two key bottlenecks limiting their performance in text-to-image generation and instruction-based editing. To address this, we propose a Pixel–Semantic VAE (`PS-VAE`) that maps representation features and pixel details into a compact, KL-regularized latent space by fine-tuning pretrained representation encoders under both pixel-level and semantic reconstruction objectives. As a result, `PS-VAE` achieves state-of-the-art performance in reconstruction, generation, and image editing. This work offers a path toward unifying visual understanding and generation in a single encoder.

**Limitations and Future Work** This study focuses on controlled comparisons under ImageNet-based reconstruction training and 256-resolution generation. Future work includes scaling `PS-VAE` to broader training datasets, evaluating its understanding capability on more comprehensive benchmarks, and extending the framework to higher-resolution generation. Our preliminary DINOv3-based 512-resolution results already show encouraging advantages over Flux-VAE in generation performance, suggesting that native-resolution representation encoders are a promising direction for high-resolution `PS-VAE`.

## Acknowledgments

This work is partially supported by the General Research Fund of Hong Kong No.17208825, 17200622 and 17209324.

## Impact Statement

This paper aims to advance visual generation research by improving how variational autoencoders (VAEs) represent visual information for generative modeling. By studying how representation encoders can be adapted for generative tasks, this work contributes to more accurate and efficient modeling of visual data, which may benefit a broad range of downstream applications, including image generation, image editing, and unified vision understanding and generation.

As a methodological contribution focused on training paradigms and architectural design, this work does not introduce application-specific systems or new datasets, nor does it directly target sensitive domains involving personal data, decision-making about individuals, or automated social control. The proposed techniques follow standard practices in machine learning research and are intended for general-purpose use in advancing generative and representation learning.

While advances in generative modeling may potentially be misused to produce misleading or harmful content, such risks are common to generative models in general and are not unique to the methods proposed in this paper. Addressing these broader societal risks requires system-level safeguards, responsible deployment practices, and policy considerations that are beyond the scope of this work. We therefore believe that this research does not introduce new or unique ethical concerns beyond those already well recognized in the machine learning community.

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

# A. Appendix

This appendix supplements the main paper with additional implementation details, extended experimental analyses, and qualitative results. The contents are organized as follows:

- **Generation Architecture.** We provide an ablation study of deep-fusion generation architectures (Section A.1).

- **Training and Evaluation Details.** We report training setups and evaluation protocols for reconstruction, text-to-image generation, and instruction-based editing (Section A.2).

- **Qualitative Editing Results.** Figure 12 presents visual editing examples across different feature spaces, referenced in the main text to highlight the role of reconstruction fidelity in preserving visual consistency.

- **FID/gFID vs. Human Perception.** We discuss why FID-based metrics can be misaligned with human perception under common evaluation settings (Section A.3).

- **Toy Example for Off-Manifold Behavior.** We provide a toy experiment that validates the off-manifold analysis (Section A.4).

- **Loss Details of S-VAE and PS-VAE.** We specify the loss components and training objectives used in S-VAE and PS-VAE, and qualitatively discuss the effect of balancing semantic and pixel reconstruction losses (Section A.5).

## A.1. Generation Architecture

Unified models for generation and understanding are being actively explored. Owing to its strong semantic representation and high-fidelity reconstruction, PS-VAE has strong potential to serve as a unified encoder in such frameworks. For these reasons, we adopt a deep-fusion architecture as our generation paradigm. To investigate which deep-fusion architecture yields superior performance, we first conduct a preliminary ablation study on three popular deep-fusion designs for generation, as illustrated in Figure 11. (a) LlamaFusion (Shi et al., 2024),

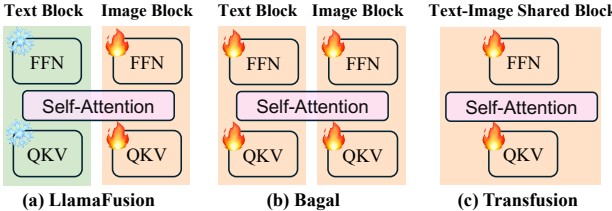

*Figure 11.* Block comparison of three deep-fusion architectures.

which freezes all language blocks and adds parallel image blocks with identical architecture; (b) Bagel-style models (Deng et al., 2025), which unfreeze both text and image branches to improve multimodal alignment; and (c) Transfusion (Zhou et al., 2025), which processes image and text tokens jointly using fully shared transformer blocks.

We evaluate the three deep-fusion architectures using VAVAE (Li et al., 2024) (32-channel latent, stride-16, patch size 1). All fusion blocks are initialized from Qwen2.5-0.5B (Bai et al., 2023). We apply 2D positional encoding to the VAE features and inject timestep embeddings into their initial hidden states before feeding them into the LLM backbone, following Bagel (Deng et al., 2025). For text-to-image, we concatenate text em-

*Table 6.* GenEval scores of Deep-Fusion architectures.

| Model | Params (M) | GenEval ↑ |
|---|---|---|
| LlamaFusion | 857 | 0.524 |
| Bagel | 857 | 0.763 |
| Transfusion | 500 | 0.752 |
| Transfusion + Wide DDT Head | 653 | 0.762 |

beddings with the noisy image latent. Text tokens use a causal mask, while the noisy image latent uses full attention mask. For image editing, we concatenate the clean latents of input images, the instruction text embeddings, and the noisy latents. We apply a full attention mask to the clean and noisy latents, while employing a causal mask for the instruction text.

Results in Table 6 indicate that LlamaFusion exhibits a clear bottleneck, likely due to its frozen language branch being unable to adapt to text-to-image generation. Compared to the Transfusion-style block design, the Bagel-style design improves performance by 1.1 but increases parameters by 71%. Since we only evaluate text-to-image performance across different feature spaces and do not consider preserving language modeling capability, we adopt the Transfusion-style block as our core fusion architecture for better parameter efficiency.

With the fusion block fixed, we incorporate the wide DDT head (Wang et al., 2025) from RAE (Zheng et al., 2025), which enhances generation quality in high-channel feature spaces (as we analyzed in Section 3.2). We validate its effectiveness

through consistent gains across multiple VAEs. As shown in Table 6, the head improves VAVAE (32-channel, stride-16, patch size 1) from 75.2 to 76.2. We observe similar improvements for Flux-VAE (Labs, 2024) (16-channel, stride-8, patch size 2), which increases from 63.7 to 68.04, and MAR-VAE (16-channel, stride-16, patch size 1), which rises from 72.6 to 75.75. Given these consistent results, we adopt the wide DDT head as a standard component.

### A.2. Training and Evaluation Details

**Reconstruction** To ensure a fair comparison with prior work, we train our reconstruction models exclusively on ImageNet-1K (Russakovsky et al., 2015), though we note that future work could benefit from larger, more diverse datasets. Input images are resized and center-cropped to $224 \times 224$. Using a patch size of 14 results in a sequence length of $16 \times 16$, making the computation—in terms of both FLOPs and runtime—significantly more efficient than the VAE-style encoders used in Latent Diffusion Models (LDMs) (Rombach et al., 2022). Our pixel decoder adopts the LDM architecture (Rombach et al., 2022) and reconstructs images at a resolution of $256 \times 256$. We evaluate performance using rFID, SSIM, PSNR, and LPIPS on the ImageNet-1K validation set. Models are trained with a batch size of 96 and a learning rate of $10^{-4}$. We employ a two-stage training strategy: first, we freeze the foundation model and train only the semantic encoder and decoder, with the pixel decoder trained on detached semantic latents to prevent interference with semantic compression. Training is conducted for approximately 300K steps. In the second stage, we unfreeze all components, allowing gradients from the pixel decoder to backpropagate to both the foundation model and the semantic encoder. The loss weights for $\mathcal{L}_S$ and $\mathcal{L}_P$ are set to 0.1 and 1, respectively. This training stage runs for approximately 500K steps.

**Text-to-Image Generation** We utilize CC12M-LLaVA-NeXT (Changpinyo et al., 2021; Emporium, 2024) for training, which comprises 10.9 million images with detailed long-form captions (Liu et al., 2024a). Images are resized and center-cropped to $256 \times 256$. We evaluate performance using GenEval (Ghosh et al., 2023) and DPG-Bench (Hu et al., 2024). GenEval relies on object detection, making it highly sensitive to structure and texture—factors closely tied to human perceptual preference. If generated objects exhibit geometric inaccuracies or distorted textures, the detector may fail to classify them correctly or produce duplicate detections. As a result, scores can be lower even when the text–image semantic alignment appears correct at a glance. Conversely, DPG-Bench employs a vision–language model as a judge, prioritizing high-level alignment over fine-grained details. This complementarity allows us to better interpret trade-offs between structural fidelity and semantic alignment. We train with a batch size of approximately 730, a learning rate of $10^{-4}$, and apply EMA with a decay of 0.9999. Training for 200K iterations ensures convergence across various generative feature spaces. For GenEval, we use the rewritten long-prompt version from Bagel (Deng et al., 2025), consistent with our long-caption training data.

Variations in patch size and channel dimensionality along the sequence length alter the signal-to-noise ratio (SNR) during interpolation between noise and latents. To maintain consistent SNR weighting across feature spaces, we apply a shifted timestep $t' = \frac{shift\_factor \cdot t}{1+(shift\_factor-1) \cdot t}$, where $t$ is sampled from a Logit-Normal distribution (Esser et al., 2024; Zheng et al., 2025). Since the sequence length is fixed across feature spaces, the shift factor depends only on the latent channel dimension $C_{\text{vae}}$ and patch size $P_{\text{vae}}$: $shift\_factor = \sqrt{\frac{C_{\text{vae}} P_{\text{vae}}^2}{C_{\text{base}} P_{\text{base}}^2}}$, where $C_{\text{base}} = 16$ and $P_{\text{base}} = 1$. For instance, Flux (Labs, 2024) ($C = 16, P = 2$) yields a shift factor of 2, while DINOv2-B ($C = 768, P = 1$) yields approximately 6.93. We conduct ablation studies to confirm that these calculated values are reasonable. For Flux, a factor of 2 yields a better GenEval score compared to factors of 1 and 3, while for RAE, a factor of 6.93 performs best compared to 6 and 8. We therefore apply this rule to all feature-space and channel-number ablation experiments. During inference, we use 50-step Euler sampling with a timestep shift of 3 and a classifier-free guidance scale of 6.5.

### A.3. FID $\neq$ Human Perception

We observe a systematic mismatch between FID-based metrics and human perception. In class-to-image studies, visual demos are often sampled with very high classifier-free guidance (Ho & Salimans, 2022) (CFG), resulting in images that appear visually convincing to human observers. However, FID is typically reported at low CFG, where better scores are easier to obtain, even though the resulting images often exhibit severe structural errors and appear poor to humans. Since FID is largely insensitive to structural correctness, many severe artifacts exhibited by RAE are obscured under FID-based evaluation protocols.

In this sense, FID used in class-to-image experiments should be viewed as a practical compromise under limited training budgets, rather than a definitive measure of generative quality. With sufficient compute, we encourage the community to adopt more comprehensive, human-aligned evaluation protocols for text-to-image generation to enable more reliable

| Input | MAR-VAE | VAVAE | RAE | Ours |

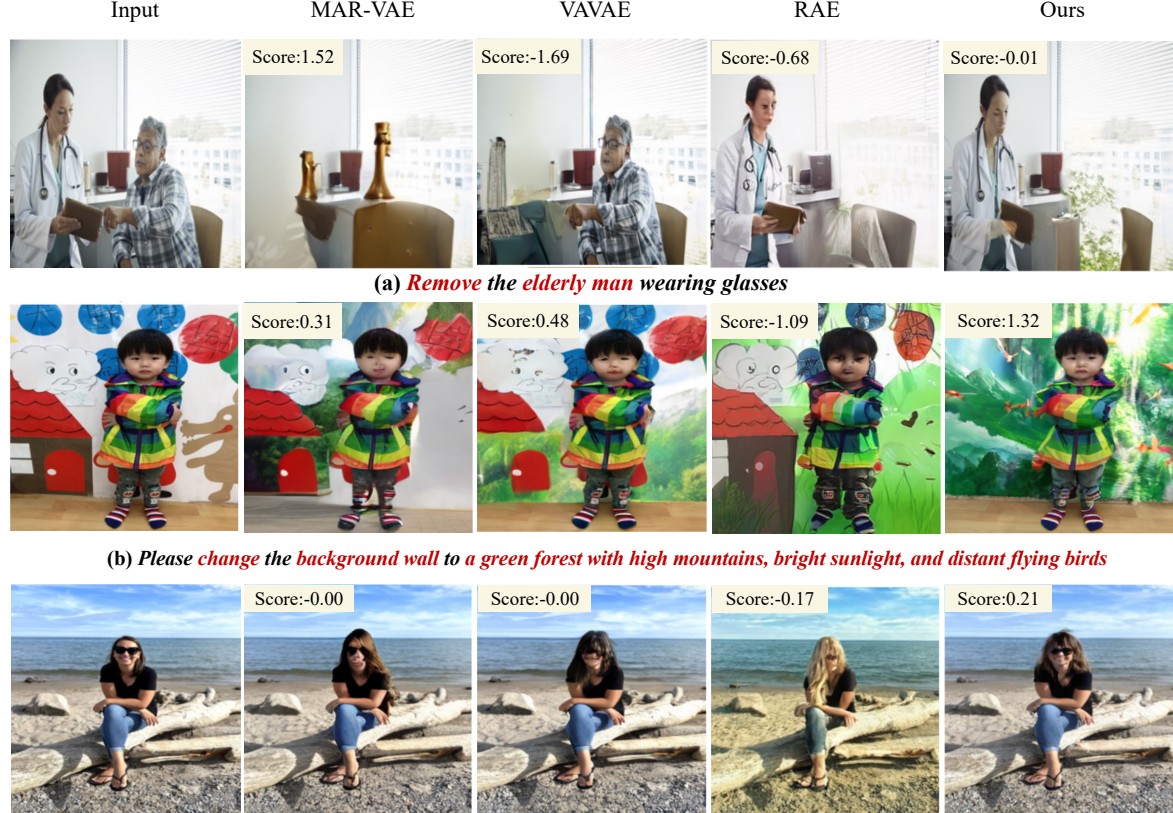

**(a)** *Remove the elderly man wearing glasses*

**(b)** *Please change the background wall to a green forest with high mountains, bright sunlight, and distant flying birds*

**(c)** *Add more hair to the front, making it long and soft for a gentle look*

*Figure 12.* **Editing visual examples of models trained on different feature spaces.** As shown in (a), both `PS-VAE` and RAE exhibit reasonable visual grounding, correctly identifying the elderly man and the background wall in (a). However, RAE's performance is strongly limited by its weak reconstruction ability, resulting in details inconsistent with the input image (a,b,c). In contrast, benefiting from strong semantic alignment and high-fidelity reconstruction, `PS-VAE` achieves accurate instruction following while preserving consistent visual details between the input and edited images, such as the human face in (a,b,c).

validation.

## A.4. Understanding Off-Manifold Behavior via a Toy Example

To validate the theoretical analysis, we investigate diffusion training in a high-dimensional feature space (h = 8) that implicitly contains a lower-dimensional intrinsic manifold (l = 2). We construct a ground-truth 2D "PS"-shaped distribution $z$ and embed it into $\mathbb{R}^8$ via a linear isometric mapping $x = Qz$, where $Q \in \mathbb{R}^{8 \times 2}$ has orthonormal columns. In this setup, $Q$ acts as the linear decoder defining the manifold. We then train identical 256-channel MLP-based diffusion models separately on the intrinsic 2D data and the embedded 8D data. For evaluation, we project the generated 8D samples back to the 2D plane using the linear encoder $Q^\top$ (noting that $Q^\top Q = I_2$, which perfectly recovers on-manifold data). As shown in Figure 3, learning in the 8D ambient space results in slower convergence and a degradation in sample quality. Nearest-neighbor distance evaluation against the ground truth reveals that the top 5% tail samples from the 8D model deviate significantly from the true manifold. Despite sharing the same intrinsic geometry, the unconstrained high-dimensional representation amplifies off-manifold behavior. This confirms that discovering and training on the intrinsic low-dimensional isomorphic distribution is essential for stabilizing diffusion training and eliminating generation artifacts.

## A.5. Loss Details of S-VAE and PS-VAE

We first detail the loss components used in the semantic reconstruction loss $\mathcal{L}_S$ and the pixel reconstruction loss $\mathcal{L}_P$.

**Semantic reconstruction loss** The semantic reconstruction loss $\mathcal{L}_S$ is designed to align two high-dimensional semantic feature representations, namely the original feature $f'_h$ extracted from the frozen representation encoder and the reconstructed feature $f''_h$ produced by the semantic decoder. It consists of three complementary terms: an $\ell_2$ loss $\mathcal{L}_{\ell_2}$, a cosine similarity

loss $\mathcal{L}_{\text{cos}}$, and a matrix similarity loss $\mathcal{L}_{\text{mat}}$.

Let $f'_h, f''_h \in \mathbb{R}^{H \times W \times C}$ denote the semantic feature maps, and let $N = H \times W$ be the total number of spatial locations. We flatten the feature maps along the spatial dimension and denote the feature vector at location $i$ as $f'_{h,i}$ and $f''_{h,i}$, respectively. The $\ell_2$ loss enforces point-wise reconstruction of feature magnitudes and is defined as

$$\mathcal{L}_{\ell_2} = \frac{1}{NC} \sum_{i=1}^{N} \left\| f'_{h,i} - f''_{h,i} \right\|_2^2. \tag{3}$$

To further align the directional semantics of the features and reduce sensitivity to feature scale, we adopt a cosine similarity loss,

$$\mathcal{L}_{\text{cos}} = \frac{1}{N} \sum_{i=1}^{N} \left( 1 - \frac{\langle f'_{h,i}, f''_{h,i} \rangle}{\| f'_{h,i} \|_2 \, \| f''_{h,i} \|_2} \right). \tag{4}$$

While the above two terms focus on point-wise alignment, they do not explicitly constrain the relative structure across spatial locations. To preserve the spatial relational structure of the feature maps, we introduce a matrix similarity loss following (Yao et al., 2025). This loss aligns the pairwise cosine similarity matrices of $f'_h$ and $f''_h$ and is defined as

$$\mathcal{L}_{\text{mat}} = \frac{1}{N^2} \sum_{i,j} \left( \left| \frac{\langle f'_{h,i}, f'_{h,j} \rangle}{\| f'_{h,i} \|_2 \, \| f'_{h,j} \|_2} - \frac{\langle f''_{h,i}, f''_{h,j} \rangle}{\| f''_{h,i} \|_2 \, \| f''_{h,j} \|_2} \right| \right) \tag{5}$$

.

The overall semantic reconstruction loss is given by

$$\mathcal{L}_S = \mathcal{L}_{\ell_2} + \mathcal{L}_{\text{cos}} + \mathcal{L}_{\text{mat}}. \tag{6}$$

**Pixel reconstruction loss** The pixel reconstruction loss $\mathcal{L}_P$ is used to reconstruct the input image $I_{\text{input}}$ from the latent representation via the pixel decoder. Following common practice in image generation and reconstruction (Rombach et al., 2022), $\mathcal{L}_P$ consists of an $\ell_1$ reconstruction loss, a perceptual loss, and an adversarial loss. Specifically, it is defined as

$$\mathcal{L}_P = \mathcal{L}_{\ell_1} + \lambda_{\text{perc}} \mathcal{L}_{\text{perc}} + \lambda_{\text{gan}}^{\text{adapt}} \mathcal{L}_{\text{gan}}, \tag{7}$$

where $\mathcal{L}_{\ell_1} = \| I_{\text{input}} - I_{\text{output}} \|_1$, $\mathcal{L}_{\text{perc}}$ denotes the perceptual loss computed on pretrained VGG features, and $\mathcal{L}_{\text{gan}}$ is an adversarial loss that encourages photorealistic image reconstruction.

Following prior studies (Rombach et al., 2022), the adversarial loss is weighted using a gradient-norm–based adaptive scheme to balance reconstruction and adversarial objectives during training. In practice, we adopt a fixed scaling factor of 0.5 for the adaptive adversarial weight. Unless otherwise specified, we set $\lambda_{\text{perc}} = 1.0$ and the $\ell_1$ loss weight to 1.0.

Following prior studies (Zheng et al., 2025; Ma et al., 2025), we further introduce an engineering refinement that uses a frozen DINOv2 (Oquab et al., 2023) encoder with a lightweight trainable head as the discriminator, which significantly improves training stability.

**KL Regularization of Latent Space** Latent diffusion models (Rombach et al., 2022) employ a Kullback–Leibler (KL) divergence term to regularize the latent distribution produced by the encoder. Specifically, the encoder is trained to map the input to a Gaussian posterior distribution $q(z \mid x) = \mathcal{N}(\mu, \sigma^2)$, which is encouraged to match a standard normal prior $\mathcal{N}(0, I)$ via a KL divergence loss. This regularization enforces a compact and well-behaved latent space. Formally, the KL loss is defined as

$$\mathcal{L}_{\text{KL}} = D_{\text{KL}}\big( q(z \mid x) \,\|\, \mathcal{N}(0, I) \big), \tag{8}$$

which admits a closed-form expression for diagonal Gaussian posteriors.

**S-VAE Stage** During the S-VAE stage, the representation encoder is frozen; therefore, the extracted feature $f'_h$ is identical to the original encoder feature $f_h$, and we use $f'_h$ as the reconstruction target. The overall objective in this stage is

$$\mathcal{L} = \mathcal{L}_S(f'_h, f''_h) + \lambda_P \mathcal{L}_P(I_i, I_o) + \beta \mathcal{L}_{\text{KL}}, \qquad \beta = 1 \times 10^{-6}, \tag{9}$$

where the pixel reconstruction loss $\mathcal{L}_P$ is computed from the detached latent $f_l.\text{detach}()$. As a result, $\mathcal{L}_P$ only updates the pixel decoder and does not backpropagate gradients to the representation encoder or the semantic VAE. Therefore, the relative weighting between $\mathcal{L}_S$ and $\mathcal{L}_P$ is not critical in this stage.

**PS-VAE Stage** During the PS-VAE stage, we unfreeze the representation encoder and allow it to be jointly optimized with the semantic and pixel decoders.

$$\mathcal{L} = \lambda_S \, \mathcal{L}_S(f_h, \, f_h', \, f_h'') + \lambda_P \, \mathcal{L}_P(I_i, \, I_o) + \beta \, \mathcal{L}_{\text{KL}}(f_l) . \tag{10}$$

In this stage, the semantic reconstruction loss $\mathcal{L}_S$ is applied to align the intermediate feature $f_h'$ and the reconstructed feature $f_h''$ with the original frozen representation feature $f_h$, ensuring that semantic consistency is preserved. Meanwhile, the pixel reconstruction loss $\mathcal{L}_P$ is allowed to backpropagate gradients into the representation encoder, encouraging it to capture fine-grained visual details required for high-fidelity image reconstruction. To balance semantic preservation and pixel-level detail learning, we set $\lambda_S = 0.1$ and $\lambda_P = 1.0$. Qualitatively, increasing the relative weight of $\mathcal{L}_S$ better preserves semantic structure but can slow detail enrichment, whereas overly weak semantic supervision biases the model toward pixel reconstruction and degrades generation behavior, approaching the pixel-only P-VAE variant. The KL regularization weight is kept as $\beta = 1 \times 10^{-6}$ to maintain a compact and well-behaved latent space throughout training.

