# OpenReview forum: "Both Semantics and Reconstruction Matter: Making Representation Encoders Ready for Text-to-Image Generation and Editing"
_ICML.cc/2026/Conference — ICML 2026 regular_

### Official Review · Reviewer_KhPH · 2026-02-20

**Soundness:** 2
**Presentation:** 3
**Significance:** 3
**Originality:** 2
**Overall Recommendation:** 4
**Confidence:** 4

**Summary:**

PS-VAE proposes a two-stage training approach to adapt representation encoders such as DINOv2 for generative tasks. In the first stage, the encoder is frozen and a semantic autoencoder is trained to compress high-dimensional features into a compact 96-dimensional space with KL divergence regularization, addressing the off-manifold generation problem of diffusion models in high-dimensional spaces. In the second stage, all components are unfrozen and jointly optimized with pixel reconstruction loss (forcing the encoder to preserve high-frequency details) and semantic distillation loss (supervising the fine-tuning process with the frozen original encoder to prevent semantic degradation), ultimately constructing a compact latent space that is both semantically rich and reconstruction-faithful. The method is trained for reconstruction on ImageNet-1K and for text-to-image generation on CC12M. Experiments further validate that the framework transfers to SigLIP2 encoders, and the 96-channel variant continues to improve as the generative model scales.

**Compliance With Llm Reviewing Policy:**

Affirmed.

**Final Justification:**

The authors provided supplementary evaluations on understanding capabilities, validating the feasibility of the proposed method.

**Key Questions For Authors:**

1.	What irreplaceable benefits does the increased computational cost provide compared to RAE?
2.	Could the authors provide evaluation results on more diverse multimodal understanding benchmarks, such as MMStar, MMBench, and VideoMME, to more comprehensively verify whether the compressed latent space compromises the encoder's understanding capabilities?
3.	Does compressing high-dimensional semantic features into a 96-dimensional space impose a bottleneck on performance upper bounds for complex understanding tasks?

**Limitations:**

Yes

**Strengths And Weaknesses:**

-	Strengths
1.	By introducing a 96-dimensional compact bottleneck, PS-VAE forcibly regularizes the distribution, fundamentally resolving the object structure distortions commonly seen when diffusing over sparse, high-dimensional features. It further demonstrates that unfreezing the encoder and incorporating pixel-level losses can recover the high-frequency details missing in representation encoders without significantly harming semantics.
2.	The architecture is not only effective on DINOv2 but also, through transfer experiments to SigLIP2, proves that this two-stage training framework serves as a general-purpose "representation-to-generation" protocol.
-	Weaknesses
1.	The novelty of this paper is somewhat limited. The motivation shares similarities with FAE [1], and the overall framework appears to be an incremental optimization of RAE. This improvement comes at the cost of increased computational complexity and higher training overhead.
2.	The evaluation of understanding capabilities is insufficient. Changes in understanding metrics are not presented in a structured manner; only two benchmarks are briefly mentioned in the main text, which is relatively one-sided. At a minimum, it should include two benchmarks for image understanding and two for video understanding (e.g., MMStar, MMBench, MME-RealWorld, VideoMME, MVBench, MVLU). Additionally, the dimensionality reduction could potentially impact the upper bound of training for understanding tasks, a point that warrants more careful consideration.
[1] One Layer Is Enough: Adapting Pretrained Visual Encoders for Image Generation, 2025.

---

> ### Author Rebuttal · Authors · 2026-03-31
>
> Thank you for the careful review and for your encouraging positive feedback. We are especially encouraged that you recognized both the core paradigm of PS-VAE and the value of our experimental validation. Below, we address the main weaknesses (W1, W2) and key questions (Q1, Q2, Q3).
>
> ---
>
> ### W1 \& Q1. Novelty (relative to FAE) and irreplaceable benefits compared with RAE.
>
> **Novelty (relative to FAE)**
>
> Thank you for pointing out FAE, which is a solid work. After checking carefully, we note that **our release times were separated by only 11 days. As a concurrent work, FAE does not reduce the novelty of our contribution**.
>
> **From a methodological perspective, FAE is more closely aligned with our S-VAE stage, which is only one part of our full method**. In comparison, our work more explicitly identifies and analyzes the need to make the latent space compact, and, more importantly, we study the problem in the more practical and challenging settings of T2I and editing rather than C2I settings. This allows us to further observe that weak reconstruction is another major bottleneck. This is exactly what motivates us to go one step further from S-VAE to PS-VAE. In this sense, our contribution is not only compactifying the representation latent, but also showing that both semantic compactness and reconstruction fidelity are necessary for making representation encoders truly usable for T2I and editing. **We will add these discussion in the revision.**
>
> **Irreplaceable benefits compared with RAE**
>
> Compared with RAE, our gains come from **a fundamental improvement in the training paradigm**, rather than simply additional tuning. RAE suffers from two core limitations: a non-compact feature space and weak reconstruction, which make it intrinsically difficult to become a practically strong solution for T2I (inaccurate object structure and texture) and editing (weak consistency). **In our experiments, these bottlenecks are not resolved by the original RAE-style paradigm alone, even with longer training budgets**. In contrast, our training paradigm directly addresses both issues, with only **negligible additional overhead, since the semantic encoder is highly lightweight and consists of only 3 Transformer blocks**. More importantly, VAE training cost is negligible in a generation system compared with diffusion training.
>
> ---
>
> ### W2 & Q2. Understanding Evaluation is Insufficient
>
> We would like to clarify that the **focus of this study is generation and editing, as also reflected in our title**. The discussion of understanding performance is intentionally brief and only meant to suggest that our method may also be useful for unified models.
>
> We additionally report results on more image understanding benchmarks (and apologize for a typo: VBench should be MMBench). The changes are small: MME-P drops from 1685 to 1652, MMBench from 85.0 to 84.7, MMMU from 0.53 to 0.52, and MMVP from 0.70 to 0.69. Notably, these results are obtained without re-training bagel, as noted in L370–L376. With proper re-training, these small drops may disappear and performance could even improve due to the improved reconstruction quality.
>
> For video understanding benchmarks, it is difficult to provide meaningful comparisons because the original Bagel model is not trained or evaluated for video understanding. We therefore believe such benchmarks are better left to future work, given the scope of this paper. We thank the reviewer for this valuable suggestion.
>
> ---
>
> ### Q3. 96-dimensional space for understanding tasks
>
> First, this is a valuable question, though a rigorous answer is beyond the scope of this paper because our focus is on generation and editing. Based on our experiments, around 100 latent channels already seem sufficient to preserve the original semantic features for image understanding, likely due to the sparsity of the original features. A concurrent work, VTP [1], also reports a similar observation.
>
> Second, we think this question may also reflect a difference in perspective on how unified models should be built. Under a Transfusion-style architecture, the reviewer’s concern is indeed critical. However, our view is closer to Bagel-style training, where we find that generation features from the final diffusion step are not ideal for understanding. Instead, understanding features should rely on an additional computation in LLM rather than directly reusing generation features. In this architecture, the understanding feature computation can still use high-channel features. From this perspective, our method mainly avoids duplicate image encoding during training while also reducing the semantic gap between generation and understanding inside the LLM blocks. Unified models are still evolving rapidly, and this is only our current view.
>
>
> [1] Towards Scalable Pre-training of Visual Tokenizers for Generation

---

> > ### Author Rebuttal · Reviewer_KhPH · 2026-04-03
> >
> > The authors provided supplementary evaluations on understanding capabilities, validating the feasibility of the proposed method.

---

> > > ### Author Response · Authors · 2026-04-03
> > >
> > > Thank you for your thoughtful follow-up. We also sincerely appreciate your valuable feedback on our work. In future work, we plan to extend this line of research beyond generation toward a more unified encoder framework, and further investigate whether PS-VAE can serve as a strong unified encoder.

---

### Official Review · Reviewer_o7f6 · 2026-02-21

**Soundness:** 3
**Presentation:** 3
**Significance:** 2
**Originality:** 2
**Overall Recommendation:** 4
**Confidence:** 3

**Summary:**

This paper studies how to adapt representation encoders, originally trained for visual understanding, as latent spaces for diffusion-based text-to-image generation and image editing. The authors identify two core challenges: the lack of compact regularization in discriminative feature spaces, which leads to off-manifold diffusion and structural artifacts, and weak pixel-level reconstruction, which limits fine-grained detail modeling. To address these issues, they propose a semantic–pixel joint reconstruction objective that regularizes and compresses the latent space into a compact yet semantically rich representation. The experiments on image generation and image editing are conducted.

**Compliance With Llm Reviewing Policy:**

Affirmed.

**Final Justification:**

My concerns have been addressed by author response.

**Key Questions For Authors:**

1. What is the core technical novelty beyond latent compression and standard VAE regularization?
The proposed method appears to mainly learn a projection from high-dimensional representation features to a lower-dimensional KL-regularized latent space with semantic and pixel reconstruction losses. Can the authors clearly articulate what fundamentally new algorithmic or theoretical component is introduced beyond established VAE and diffusion practices?

2. Can you provide stronger evidence or formal analysis of the “off-manifold diffusion” claim?
The current justification is largely intuitive and supported by toy examples and empirical observations. Is there any quantitative measurement of manifold proximity in real feature spaces, or theoretical analysis showing why high-dimensional representation latents necessarily degrade diffusion learning?

3. How does the method scale to higher resolutions (e.g., 512×512 or 1024×1024)?
Most experiments are conducted at 256×256 resolution. Have the authors evaluated whether the compact 96-channel latent space maintains reconstruction fidelity and generative quality at higher resolutions?

4. The method introduces semantic projection, joint reconstruction objectives, and staged training. How does training cost (FLOPs, memory usage, convergence time) compare to standard VAE latents or RAE-style approaches?

**Limitations:**

The paper does not appear to include a dedicated discussion of limitations.

--Results suggest that higher-dimensional latents may benefit from larger diffusion backbones. It remains unclear whether, with sufficiently large backbone capacity, the original high-dimensional representation space could be modeled effectively without requiring projection into a compact latent space.

--As the work improves text-to-image generation and editing quality, it may contribute to more realistic synthetic media, which can be misused for misinformation or harmful content generation.

**Strengths And Weaknesses:**

Strength:

--The paper clearly diagnoses practical issues when directly using representation encoder features as diffusion latents, namely off-manifold instability and weak pixel-level reconstruction. The motivation is well articulated and supported with empirical observations.

--The authors provide thorough ablations (RAE → S-VAE → PS-VAE, channel scaling, backbone transfer), which make the design choices transparent and reasonably well validated.

Weakness:

--The core method mainly consists of learning a projection from high-dimensional representation features to a lower-dimensional KL-regularized latent space, combined with standard semantic and pixel reconstruction losses. These components (latent compression, KL regularization, reconstruction objectives, staged training) are well established in prior VAE and diffusion literature, limiting conceptual originality. The use of KL regularization and reconstruction losses follows established VAE practice, though applied in a new context. The gains seem to stem from careful system design and tuning rather than a novel algorithmic insight, which may not justify acceptance at a top-tier venue focused on methodological breakthroughs.

--Compared to RAE-style approaches, the contribution appears to be a stabilization and refinement of latent design rather than a fundamentally new modeling framework or theoretical advancement.

--Although the paper discusses off-manifold diffusion, the analysis remains largely intuitive and does not provide rigorous theoretical guarantees or new generative principles.

--Most experiments are at 256×256 resolution. It remains unclear whether the benefits scale cleanly to high-resolution production-level systems.

--The overhead of training semantic encoders and the impact on memory/compute compared to standard VAEs is not deeply analyzed.

--The results suggest that high-dimensional latents may benefit from larger diffusion backbones. If sufficient model capacity is provided, it is possible that the original high-dimensional representation space could be modeled effectively without requiring an additional projection to a compact latent space. This raises the question of whether the proposed mapping is fundamentally necessary, or primarily a practical trade-off under limited backbone capacity.

---

> ### Author Rebuttal · Authors · 2026-03-31
>
> Thank you for the careful review and valuable questions. We appreciate that you recognized the two practical challenges we solve when using representation features as generative latents, and found our experiments transparent and well validated. Below, we address your main weaknesses (W) and questions (Q).
>
> ---
>
> ### W1 \& W2 \& Q1. Novelty and fundamental progress beyond RAE
>
> **Our main contribution is not a new standalone loss term, but a systematic framework that resolves the two key obstacles preventing representation encoders from serving as practical generative latents for T2I and editing**: non-compact latent geometry and weak reconstruction fidelity. More importantly, it helps bridge the long-standing gap between understanding-oriented representation encoders and generation-oriented VAEs, which has long hindered unified vision-language models. To our knowledge, **this is the first work to show, with solid T2I and editing results, that representation encoders can outperform conventional VAEs in practical generative settings.**
>
> Compared with RAE, which directly uses a frozen representation encoder, PS-VAE makes a more fundamental step by explicitly addressing two key practical gaps for T2I and editing: off-manifold generation in non-compact latent spaces, and weak reconstruction caused by understanding-oriented training objectives. We therefore believe PS-VAE goes beyond a simple refinement of RAE in the practical T2I/editing.
>
> ---
>
> ### W3 \& Q2. About off-manifold diffusion analysis
>
> As for theoretical analysis, we believe our discussion in L212--L219 provides an important explanation of why diffusion becomes more difficult in high-dimensional spaces. **It not only supports our experiments, but also helps explain the model structure proposed in RAE, as noted in L215--L216.**
>
> Regarding off-manifold diffusion in real feature spaces, **directly measuring the distance to the true image manifold is essentially difficult, since the natural image manifold is unknown and extremely complex. Modeling it is the central problem in image generation.** This is exactly why we use an example with a known ground-truth manifold: it provides a controlled setting where the off-manifold effect can be directly visualized. We think that the more obvious structural errors and worse benchmark performance of high-dimensional representation latents, compared with VAE latents, are very direct evidence of off-manifold generation.
>
>
> ---
>
> ### W4 \& Q3. Higher-resolution results
>
> As we explained in W2 and Q2 for Reviewer NiEN, **we chose the 256 setting in this study mainly for fair comparison with the baselines**. However, when combined with native-resolution representation encoders, PS-VAE can naturally extend to higher-resolution settings.
>
>
> ---
>
> ### W5 \& Q4. The overhead training cost of PS-VAE
>
> The additional semantic encoder contains only 3 Transformer blocks, and thus introduces almost negligible memory and computational overhead.
>
> As shown in the table, PS-VAE introduces only a very limited increase in FLOPs, while achieving substantial gains in both reconstruction and generation.  More importantly, **in the full training pipeline, VAE training cost is minor compared with the cost of training the downstream DiT**. In this sense, PS-VAE is not only effective but also highly economical.
>
>
> | Method | PSNR ↑ | GenEval ↑ | DPG-Bench ↑ | Editing Reward ↑ | FLOPs@256 ↓ | Training |
> |---|---:|---:|---:|---:|---:|---|
> | Flux-VAE | **32.86** | 68.04 | 78.98 | -0.271 | ~138.0 | Unknown |
> | VAVAE | 27.71 | 76.16 | 82.45 | 0.227 | ~69.2 | ~130 epoch |
> | RAE | 19.20 | 71.27 | 81.72 | 0.059 | **~22.2** | ~**16** epoch |
> | **PS-VAE-32c** | 24.53 | 76.22 | **84.25** | **0.274** | ~27.7 | ~60 epoch |
> | **PS-VAE-96c** | 28.79 | **76.56** | 83.62 | 0.222 | ~27.7 | ~60 epoch |
>
> ---
>
> ### W6. Whether compactification is necessary
>
> Our result that PS-VAE-96c outperforms PS-VAE-32c with a 1.7B generator may cause some confusion. Importantly, **this scaling trend is observed within already compact PS-VAE latents, rather than on the original frozen representation encoder, which still suffers from clear structural and reconstruction issues.**
>
> Recent concurrent work, such as SVG-T2I [1], already scales representation-space T2I diffusion to 1024$\times$1024 with a 2.6B generator, yet its paper still shows examples with clear structural errors. This is consistent with our observations and **suggests that non-compact representation latents remain difficult to model even at near product-scale model sizes.**
>
> Our appendix experiment, **Directly Enriching High-Dimensional Features Fails**, further shows that **without compactification, improving reconstruction remains difficult and the representation encoder cannot be made ready for editing**, supporting that compact mapping is fundamentally necessary rather than merely a practical trade-off.
>
> [1] SVG-T2I: Scaling Up Text-to-Image Latent Diffusion Model Without Variational Autoencoder

---

> > ### Author Rebuttal · Reviewer_o7f6 · 2026-04-03
> >
> > Thank authors for the detailed response. I would like to raise my score to weak accept.

---

> > > ### Author Response · Authors · 2026-04-03
> > >
> > > Thank you very much for your careful follow-up and for raising your score. We truly appreciate your time, consideration, and thoughtful reading of our rebuttal. We are very grateful that our response helped clarify your concerns.

---

### Official Review · Reviewer_GB9m · 2026-03-10

**Soundness:** 3
**Presentation:** 3
**Significance:** 3
**Originality:** 3
**Overall Recommendation:** 4
**Confidence:** 4

**Summary:**

Through extensive literature research on Modern Latent Diffusion Models, the author found that RAE successfully achieved generation within the representation space by redesigning the DiT architecture to handle high-dimensional features, and achieved remarkable results in the class-conditional ImageNet benchmark test. However, the author keenly identified two key issues with this architecture:
1. The compact regularization of the feature representation was insufficient, leading to the generation of potential values far from the center of the feature space. To address this issue, the author normalized the generation space: proposing the S-VAE algorithm, which maps the frozen feature representation to a compact latent space through a semantic autoencoder, subject to KL regularization.
2. The pixel-level reconstruction ability was weak, which prevented the generator from learning accurate geometric shapes and textures. To solve this problem, the author unfroze the encoder and optimized it together with a pixel-level reconstruction loss based on the input image and a semantic reconstruction loss based on the output of the original frozen pre-trained encoder. This enabled the encoder to retain fine details while computing powerful semantic representations, thus forming the final "Pixel–Semantic VAE" model.
The author demonstrated the feasibility of their method through extensive comparative and ablation experiments. PS-VAE achieved leading performance in both reconstruction generation and image editing. This work provides a practical path for integrating visual understanding and generation capabilities into a single encoder, promoting the development of fields such as visual understanding and generation.

**Compliance With Llm Reviewing Policy:**

Affirmed.

**Key Questions For Authors:**

1. Although RAE is the baseline of this paper, the abbreviation RAE was first introduced without the full name. Please check if this situation occurs elsewhere in the article.
2. The final suggestion in the introduction is not to elaborate extensively on the experimental results. Personally, I prefer to present two to three main innovations in a list format. (This is only my personal opinion.)
3. This paper references KL-regularized latent space (Rombach et al., 2022) to propose S-VAE based on RAE. However, there is no literature introduction on this point of unfreezing the encoder and optimizing pixel reconstruction and semantic reconstruction in P-VAE or PS-VAE. Was this an original idea of the author?
4. The impact statement section is beyond the main text. It is recommended to write it in the appendix when submitting the manuscript. It should be included in the main text after formal acceptance.
5. In Figure 4, different colors can be used to distinguish the data streams of the two training stages.
6. In the conclusion section, it would be best if a brief introduction to future work could be provided to inspire readers' thinking.

**Limitations:**

yes

**Strengths And Weaknesses:**

Soundness
The authors addressed the issues of insufficient compact regularization of features and weak pixel-level reconstruction ability by applying KL regularization to feature representation and unfreezing the training of the encoder, proposing the PS-VAE method. They elaborated on the specific implementation of this method in the methodology section. The authors verified the effectiveness of the method through extensive experiments.

Presentation
The submitted article is written clearly, and the overall narrative is easy to understand.

Importance
The paper addresses the related issues encountered when using high-dimensional features in the representation encoder as the generation potential tool in potential diffusion models. By solving this problem, the authors have, to a certain extent, promoted the understanding, capability, and application of machine learning in the field of text-to-image generation. The impact is mainly focused on the image generation domain. Through open-source code or further research, its influence may extend to a broader range. By studying how to adaptively adjust the representation encoder to be suitable for generation tasks, this work contributes to more accurate and efficient modeling of visual data, which is expected to benefit a wide range of downstream applications, including image generation, image editing, and unified visual understanding and generation.

originality
Based on the problems of insufficient feature-based compact regularization and weak pixel-level reconstruction ability, the author believes that generative feature representation processing is carried out in an unconstrained space and does not adopt a compact regularization method, which leads to a mismatch between the high-dimensional characteristics of the represented features and their relatively low endogenous information content; secondly, the author believes that the training objective of the representation encoder focuses on generating sufficiently discriminative features for understanding and cannot retain the specific structure and fine visual information required for generation. Based on these two new insights, a new PS-VAE method is proposed, which to some extent promotes the development of this field.

---

> ### Author Rebuttal · Authors · 2026-03-31
>
> Thank you for the positive review and constructive feedback. We appreciate that you clearly identified the two key challenges in our work: insufficient compact regularization in the representation space and weak pixel-level reconstruction ability of discriminative encoders.
>
> ---
>
> Regarding your question on whether the PS-VAE stage is our original idea (Q3): yes, the design of unfreezing the representation encoder and jointly optimizing pixel reconstruction and semantic reconstruction is our original contribution.
>
> To clarify how we arrived at this stage, our exploration proceeded as follows.
>
> RAE[1] is a recent influential work showing that pretrained representation encoders can be strong alternatives to conventional VAEs for generation, with impressive results on ImageNet class-conditional generation. Its results highlighted to us that **semantics matter**. Motivated by this, we further explored representation encoders in the more challenging text-to-image and image editing settings, and found that directly using high-dimensional representation features still struggles to generate precise structures. As you mentioned, this inspired our S-VAE stage through the idea of a KL-regularized latent space (Rombach et al., 2022).
>
> However, we further observed that, especially in the later stage of T2I training with S-VAE, the generated textures and fine structures remain worse than those of strong VAE baselines, and editing results often suffer from poor detail consistency. This made us realize that **reconstruction also matters**. Once both properties become important, the PS-VAE stage becomes a natural next step: we unfreeze the encoder to recover generation-critical details, while preserving its semantic structure through semantic reconstruction supervision. To the best of our knowledge, we are not aware of prior work that explicitly trains a representation-based VAE in this way.
>
> ---
>
> We also thank you for the helpful presentation suggestions. We will:
> (1) spell out ``RAE'' at its first occurrence and carefully check for similar abbreviation issues;
> (2) revise the introduction to present the 2--3 main contributions more explicitly in a list format, while reducing excessive discussion of experimental results there. In the current version, we prioritized limited main-text space for technical and empirical details, but we will revise the introduction to present the main contributions more clearly in list form in the final version.
> (3) improve Figure 4 by using clearer visual distinctions (e.g., different colors) for the two training stages;
> (4) add a brief future-work discussion in the conclusion, including directions such as scaling to higher resolutions, broader datasets, and stronger generative backbones;
> and (5) follow the venue formatting requirements regarding the placement of the impact statement.
>
>
> [1] Diffusion Transformers with Representation Autoencoders

---

> > ### Author Rebuttal · Reviewer_GB9m · 2026-04-02
> >
> > I have no more questions about the author's refutation and reply.

---

> > > ### Author Response · Authors · 2026-04-03
> > >
> > > Thank you very much for your careful review and thoughtful feedback. We are glad that our rebuttal resolved your concerns, and we sincerely appreciate your support.

---

### Official Review · Reviewer_NiEN · 2026-03-12

**Soundness:** 3
**Presentation:** 3
**Significance:** 3
**Originality:** 3
**Overall Recommendation:** 4
**Confidence:** 4

**Summary:**

The paper proposes a novel framework, Pixel-Semantic VAE (PS-VAE), to adapt pre-trained representation encoders (such as DINOv2 and SigLIP2) for text-to-image generation and image editing tasks. The authors identify two primary bottlenecks when using high-dimensional discriminative features as generative latents: unconstrained feature spaces that lead to off-manifold latents with structural artifacts, and weak pixel-level reconstruction objectives that hinder the generation of fine-grained geometry and texture. To solve this, the authors introduce a two-stage approach. First, the S-VAE stage compresses the high-dimensional representation into a compact, KL-regularized latent space (e.g., 96 channels) using a semantic reconstruction objective. Next, the PS-VAE stage unfreezes the representation encoder and jointly trains it with a pixel decoder to balance both semantic preservation and pixel-level details. Ultimately, the resulting model achieves state-of-the-art reconstruction and generation performance across standard benchmarks compared to existing VAE and RAE baselines.

**Compliance With Llm Reviewing Policy:**

Affirmed.

**Final Justification:**

The paper identifies two practical obstacles (non-compact latent geometry and weak pixel-level reconstruction) when repurposing discriminative representation encoders as diffusion latents, and proposes a clean two-stage framework to address both. The progressive ablations are convincing, and the method generalizes across encoders (DINOv2, SigLIP2). My initial concerns regarding architecture entanglement and limited resolution/dataset scope have been adequately addressed in the rebuttal: the controlled comparison under a shared generation backbone isolates the latent-space contribution, and the preliminary 512-resolution results with DINOv3 offer encouraging evidence of scalability. I maintain my score of weak accept.

**Key Questions For Authors:**

1. In the PS-VAE training stage, how sensitive is the model to the weighting between the semantic reconstruction loss ($\lambda_S$) and the pixel reconstruction loss ($\lambda_P$)?

2. The generation results are highly impressive at $256 \times 256$ resolution. Have you encountered any fundamental bottlenecks or scaling laws when attempting to scale this compact 96-channel PS-VAE to higher resolutions (e.g., $512 \times 512$ or $1024 \times 1024$)?

**Limitations:**

**Yes.** The authors have adequately discussed the limitations regarding resolution limits (noting that scaling to higher resolutions is left for future work) and the reliance on ImageNet-1K for reconstruction.

**Strengths And Weaknesses:**

### Strengths:

1. The authors provide a compelling analysis of why direct diffusion in high-dimensional representation spaces fails. The toy experiment embedding a 2D manifold into an 8D ambient space clearly illustrates the off-manifold drift issue.
2. The paper thoroughly evaluates the proposed PS-VAE across multiple dimensions, including image reconstruction (ImageNet-1K), text-to-image generation (GenEval, DPG-Bench), and instruction-based editing (EditingReward). It also proves the method's generalizability by successfully applying it to SigLIP2.
3. The progressive ablation from RAE to S-VAE to PS-VAE effectively demonstrates the necessity of each proposed component. Furthermore, the ablation showing that direct high-dimensional enrichment leads to shortcut reconstruction is particularly insightful.

### Weaknesses:

1. The generation architecture relies heavily on specific choices, such as the Transfusion-style blocks and the wide DDT head. While the authors provide ablations for these, it makes it slightly challenging to isolate the exact performance gain attributable *solely* to the PS-VAE latent space versus the optimized transformer backbone.
2. **Resolution and Dataset Scope:** The text-to-image generation models are trained and evaluated primarily at a $256 \times 256$ resolution. Additionally, the reconstruction model is trained exclusively on ImageNet-1K, which may limit its open-world reconstruction capabilities compared to VAEs trained on broader datasets.

---

> ### Author Rebuttal · Authors · 2026-03-31
>
> Thank you for the positive review and for recognizing the strengths of our work, especially the off-manifold analysis, the comprehensive multi-task evaluation, and the progressive ablations of PS-VAE. Below, we respond to the main weaknesses (W1, W2) and key questions (Q1, Q2).
>
> ---
>
> ### W1. Architecture choice: Transfusion-style blocks and the wide DDT head.
> We thank the reviewer for this important point. We would like to emphasize that all latent/VAE variants are evaluated under the same generation framework, so the performance differences provide strong evidence for the advantage of PS-VAE itself rather than a specially optimized generator.
>
>  **Regarding Transfusion-style blocks**, we chose them for two reasons:
>
> (1) unified multimodal models are becoming increasingly important, and we believe PS-VAE is particularly valuable in this direction;
>
> (2) Transfusion does not require an additional large text encoder (e.g., T5), which keeps training efficient and makes our large-scale text-to-image and editing experiments across five latent spaces feasible under limited compute.
>
> **Regarding the wide DDT head**, it is used to mitigate the channel bias of standard DiT when comparing latent spaces with substantially different numbers of channels, as first observed in RAE, which shows that standard DiT cannot even fit a single image when the DiT width is smaller than the latent dimensionality. We provide further analysis of this issue in Footnote 2 (Lines 216–217). This structure is essential for a fair comparison across five latent spaces with different channel dimensionalities (e.g., FLUX: 64 (channel 16 + patch 2), VA-VAE: 32, MAR: 16, PS-VAE: 32/96, and RAE: 768).
>
> ---
>
> ### W2. Resolution and Dataset Scope
> **The key point of our experimental setting is fairness.**
> Except for FLUX, the compared VAEs (e.g., MAR, VA-VAE, and RAE) are all trained in a 256-resolution, ImageNet-style setting. To ensure that the performance gains come from the training paradigm rather than differences in data or resolution, our VAE is also trained on ImageNet, and all generation experiments are conducted at 256×256 under a consistent setup.
>
> Extending the generation experiments to higher resolutions beyond the VAE training resolution may cause latent drift, which would reduce the confidence of the experimental conclusions. In addition, text-to-image experiments are extremely expensive, and extending the experimental setting to 512×512 would substantially increase the overall experimental cost and turnaround time.
>
> ---
>
> ### Q1. Effect of the ratio between $\lambda_S$ and $\lambda_P$
>
> The ratio between $\lambda_S$ and $\lambda_P$ mainly affects the convergence speed of reconstruction, which is reflected in image reconstruction metrics at the same number of training iterations. For DINOv2, we find that when $\lambda_S : \lambda_P$ is in the range of 0.1 to 0.5, the generation metrics remain stable. In contrast, under the same reconstruction training budget, image reconstruction quality can differ substantially in our setting; for example, PSNR varies roughly from 28 to 26 dB when the ratio is adjusted within the 0.1 to 0.5 range. In our initial hyperparameter search, when $\lambda_S : \lambda_P < 0.1$, PSNR and SSIM can improve slightly, but generation performance begins to degrade, eventually approaching the P-VAE result reported in the paper (i.e., $\lambda_S : \lambda_P = 0$).
>
> Since image editing is particularly sensitive to detail consistency, and we also need a reasonable reconstruction experiment cycle, we choose 0.1 as the default setting.
>
>
> ---
>
> ### Q2. Scaling to higher resolutions
> Scaling to higher resolutions mainly depends on whether the representation encoder supports native-resolution inputs, so that semantic reconstruction can continue to effectively regularize the latent space at higher resolution. Recent encoders such as SigLIP2-NaViT and DINOv3 are promising in this regard.
>
> As a preliminary study during the limited rebuttal period, we implemented a  DINOv3-based PS-VAE variant. DINOv3 adopts 2D RoPE as its positional embedding and shows strong resolution extrapolation ability. Its current reconstruction results  and generation performance (up to 50K T2I iterations by the end of the rebuttal period) at 512 resolution are as follows:
>
> | model | rFID $\downarrow$ | PSNR $\uparrow$ | LPIPS $\downarrow$ | SSIM $\uparrow$ | GenEval @50K $\uparrow$ |
> |---|---:|---:|---:|---:|---:|
> | Flux | 0.049 | 35.606 | 0.03 | 0.938 | 13 |
> | VAVAE | 0.243 | 29.425 | 0.12 | 0.803 | 37 |
> | PS-VAE$_{DINOv3}$ | 0.108 | 31.074 | 0.08 | 0.861 | 51 |
>
> Overall, these early 512-resolution results appear broadly consistent with what we observed at 256 resolution.

---

> > ### Author Rebuttal · Reviewer_NiEN · 2026-04-03
> >
> > The authors have provided satisfactory responses to both weaknesses. For W1, the clarification that all latent variants share the same generation framework makes the controlled comparison convincing. For W2, the preliminary 512-resolution results with DINOv3-based PS-VAE, despite limited rebuttal-period compute, offer encouraging evidence that the approach scales beyond the 256 setting. The sensitivity analysis for Q1 is also informative. I have no further concerns and maintain my positive assessment.

---

> > > ### Author Response · Authors · 2026-04-03
> > >
> > > Thank you very much for your careful review and encouraging feedback. We are delighted that our rebuttal has adequately addressed your concerns, and we sincerely appreciate your positive assessment and support for our work

---

### Decision · Program_Chairs · 2026-04-30

**Decision:**

Accept (regular)

**Comment:**

This is one of those cases in which all the reviewers converge to a weak accept after the discussions and the rebuttal phase. The reviewers are satisfied with the answers provided and the minor issues that were originally raised by the them have been completely addressed. I like the well-motivated and thorough study of diffusion in high-dimensional representation spaces which clearly identifies key issues such as off-manifold drift and weak pixel-level reconstruction. Through insightful analysis and illustrative experiments, the authors justify the need for improved feature regularization. The proposed PS-VAE introduces compact latent regularization, KL constraints, and encoder fine-tuning to better align representation features with generative tasks. Overall, I believe the work is clearly presented, methodologically sound, and contributes meaningful insights and practical advances to representation-based image generation.